# Inhibition of itch by neurokinin 1 receptor (Tacr1) -expressing ON cells in the rostral ventromedial medulla in mice

**Taylor Follansbee[1,2]\*, Dan Domocos[3], Eileen Nguyen[4], Amanda Nguyen[1], Aristea Bountouvas[1], Lauren Velasquez[1], Mirela Iodi Carstens[1], Keiko Takanami[5], Sarah E Ross[4], Earl Carstens[1]**

[1]Department of Neurobiology, Physiology and Behavior, University of California, Davis, Davis, United States; [2]Department of Neuroscience, Johns Hopkins University, Baltimore, United States; [3]Department of Anatomy, Animal Physiology and Biophysics, University of Bucharest, Bucharest, Romania; [4]Pittsburgh Center for Pain Research and Department of Neurobiology, University of Pittsburgh, Pittsburgh, United States; [5]Department of Environmental Life Science, National Nara Women University, Nara, Japan

**Abstract** The rostral ventromedial medulla (RVM) is important in descending modulation of spinal nociceptive transmission, but it is unclear if the RVM also modulates spinal pruriceptive transmission. RVM ON cells are activated by noxious algesic and pruritic stimuli and are pronociceptive. Many RVM-spinal projection neurons express the neurokinin-1 receptor (Tacr1), and ON-cells are excited by local administration of substance P (SP). We hypothesized that Tacr1-expressing RVM ON cells exert an inhibitory effect on itch opposite to their pronociceptive action. Intramedullary microinjection of SP significantly potentiated RVM ON cells and reduced pruritogen-evoked scratching while producing mild mechanical sensitization. Chemogenetic activation of RVM Tacr1-expressing RVM neurons also reduced acute pruritogen-evoked scratching. Optotagging experiments confirmed RVM Tacr1-expressing neurons to be ON cells. We conclude that Tacr1-expressing ON cells in RVM play a significant role in the modulation of pruriceptive transmission.

\*For correspondence:
tfollansbee@ucdavis.edu

**Competing interest:** The authors declare that no competing interests exist.

## Editor's evaluation

This manuscript uses several complementary strategies to investigate and manipulate the activity of neurokinin 1 receptor-expressing neurons in the rostroventral medulla of mice in the context of itch sensation. The study delivers important insights into supraspinal itch processing and descending modulation of itch signals in the spinal cord.

## Introduction

The transmission of somatosensory information in the spinal cord is under top-down modulation and has been extensively studied in the context of pain signaling. Descending modulation of pain is reflected by phenomena such as expectation (placebo and nocebo), diffuse noxious inhibitory control (DNIC), and conditioned pain modulation (CPM) (*Bartels et al., 2018*; *Chebbi et al., 2014*; *Damien et al., 2018*; *Le Bars, 2002*; *Lockwood and Dickenson, 2020*). Descending modulation is also thought to underlie stress mediated changes in pain threshold, with acute stress inhibiting and chronic stress facilitating nociceptive transmission (*Butler and Finn, 2009*; *Fields, 2000*; *Jennings et al., 2014*; *Wager and Atlas, 2015*). Recent studies indicate that the spinal transmission of pruriceptive

information is also under descending modulatory influences, but it is not known whether itch is modulated in the same way as pain (*Agostinelli and Bassuk, 2021*; *Gao et al., 2021*; *Gao et al., 2019*; *Koga et al., 2020*; *Liu et al., 2019*; *Samineni et al., 2019*; *Wu et al., 2021*).

The rostral ventromedial medulla (RVM) contains neurons with descending projections to the spinal cord which bidirectionally modulate spinal nociceptive transmission (*Fields, 2000*; *Fields and Basbaum, 1978*; *Fields and Heinricher, 1985*; *Heinricher et al., 2009*; *Millan, 2002*; *Ossipov et al., 2014*). RVM ON cells are excited and OFF cells are inhibited by noxious stimulation just prior to a nocifensive withdrawal reflex, and respectively facilitate and inhibit spinal nociceptive transmission (*Fields, 2004*; *Fields et al., 1983*). Increased ON cell and decreased OFF cell firing are thought to contribute to the chronification of pain (*Ossipov et al., 2014*). Neutral cells do not exhibit any significant response to noxious stimulation (*Barbaro et al., 1986*). Many RVM neurons project to the spinal cord via the dorsolateral funiculus (DLF), terminating in layers I, II, and V, and are implicated in the analgesic effects of opioids. It is still unknown whether the RVM ON and OFF cells are involved in the modulation of itch transmission.

A recent study reported that neurons in the periaqueductal gray (PAG) expressing Tac1, the gene for substance P (SP), project to and make an excitatory glutamatergic connection with neurons in the RVM (*Gao et al., 2019*). Activation of these PAG neurons promotes scratching. A population of GABAergic neurons originating in the RVM synapse onto spinal neurons that express the gastrin releasing peptide receptor (GRPR) (*Liu et al., 2019*), which are considered to be essential for spinal pruriceptive transmission (*Sun and Chen, 2007*). Activation of GABAergic and inhibition of glutamatergic neurons in PAG reduced scratching behavior in both acute and chronic itch conditions (*Samineni et al., 2019*). Furthermore, pruritogens were shown to excite ON and inhibit OFF cells (*Follansbee et al., 2018*). Cervical cold block, which diminishes activity in descending modulatory pathways, decreased pruritogen-evoked activity and facilitated algogen-evoked activity of spinal cord neurons implying opposing effects of descending modulatory pathways on nociceptive and pruriceptive transmission (*Carstens et al., 2018*). The current evidence suggests that at least two classes of PAG neurons, via connections to RVM cells, exert bimodal effects on the spinal pain and itch signaling pathways. Since activation of PAG *tac1r* neurons facilitates itch this implies that activation of RVM Tacr1-expressing neurons may likewise facilitate itch. However, RVM ON cells are known to facilitate pain, and thus may have an opposing effect on itch transmission.

Here we investigated the role of Tacr1-expressing RVM neurons in itch modulation. In the present study, we used pharmacological and opto/chemogenetic methods to selectively activate RVM Tacr1-expressing cells, which we hypothesize represent a population of RVM ON cells, and assessed the effects on pruritogen-evoked scratching, and thermal and mechanical nociceptive behavior. When RVM Tacr1-expressings cells were activated, we observed that itch related behaviors were inhibited, while producing mild mechanical sensitization. We used optotagging to characterize these RVM Tacr1-expressing neurons as RVM ON cells. These results are the first to demonstrate an inhibitory effect of RVM ON cells on itch transmission, contrary to their known pronociceptive function.

## Results

### SP enhances RVM ON-cell responses to pinch

In rats, responses of RVM ON cells were potentiated by intramedullary microinjection of the Tacr1 agonist SP (*Budai et al., 2007*) and we wished to determine whether this is true in mouse. We identified RVM ON cells using in vivo extracellular electrophysiological recordings, with the criterion that noxious pinch elicited a >30% increase in firing that preceded the onset of the hindpaw withdrawal. Using a microinjection cannula attached to a recording microelectrode (*Figure 1A*), SP or saline was microinjected while recording from an ON cell in anesthetized mice. Microinjection of SP potentiated pinch-evoked responses (*Figure 1B*). Following SP microinjection, normalized responses of ON cells to repeated pinch stimuli were significantly increased (*Figure 1C*). The enhancement of responses lasted >1 hr. Histologically recovered recording sites were located within the RVM and adjacent regions of the medullary reticular formation (*Figure 1D*). These results show that, similar to rats, RVM ON cells in the mouse are potentiated following localized injection of SP.

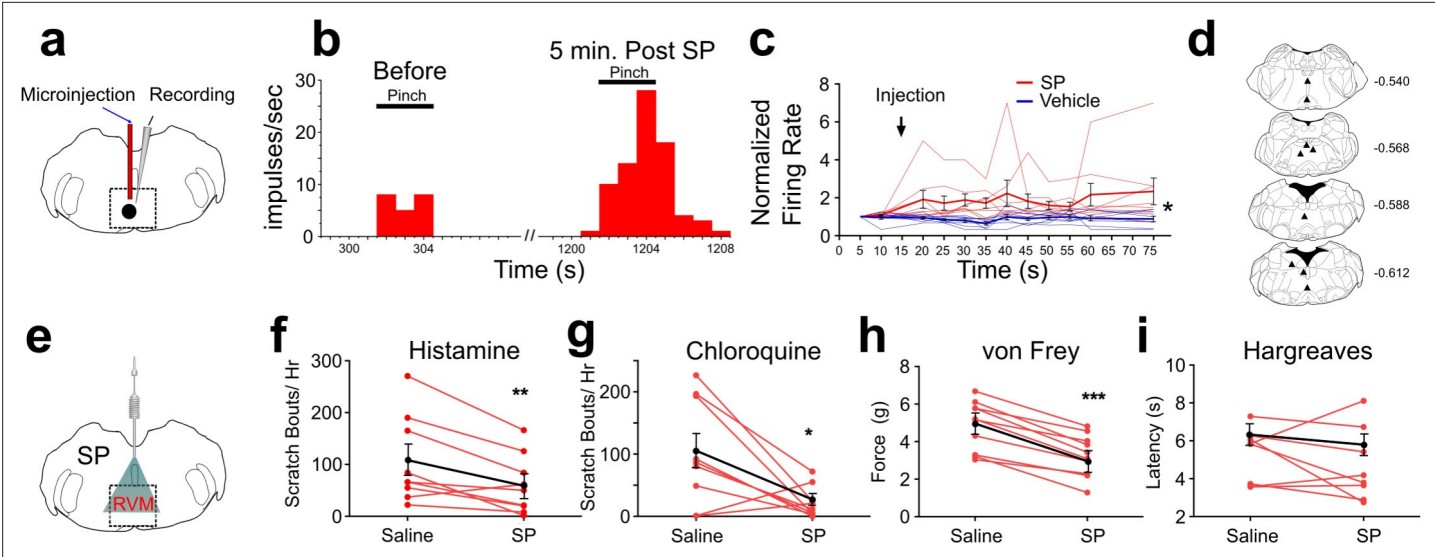

**Figure 1.** Effects of intramedullary microinjection of SP on RVM ON cells and itch and pain behavior. (**A**) SP or saline was microinjected while recording from single ON cells. (**B**) Peristimulus-time histogram of ON cell response to pinch before (left) and 5 min after local microinjection of SP (right). (**C**) Normalized firing rate of ON cells following local microinjection of saline (blue, n=8) or SP (red, n=8) at time indicated by arrow. ON cells showed a significant increase in evoked firing following SP injection compared with saline (*; two-way ANOVA, $F_{1, 14}$ = 8.020, p=0.0133, bolded lines: mean responses; error bars SEM). Male mice were used in these experiments. (**D**) Lesion sites from the RVM ON cell recordings. Numbers to right indicate Bregma coordinates. (**E**) An implanted intramedullary microinjection cannula allowed assessment of itch and pain behavior after injection of SP into RVM. (**F, G**) Graphs plot the number of scratch bouts elicited by intradermal injection of histamine (F, n=9, M = (106.2; 59.72), t=3.4113, DF = 8, 95% CI [15.06, 77.83], p=0.0092) or chloroquine (G, n=9, M = (102.9; 22.00), t=2.847, DF = 8, 95% CI [15.39, 146.5], p=0.0216) for each mouse (red dots and lines), and mean scratch bouts (black line; error bars: SEM), following intramedullary microinjection of saline or SP. Experiments with saline and SP microinjections were conducted at least 7 days apart. Microinjection of SP significantly attenuated histamine- and chloroquine-evoked scratching (**F, G**). (**H**) Mechanical withdrawal thresholds were reduced by intramedullary SP, n=10, M = (4.855; 3.253), t=8.96, DF = 8, 95% CI [1.198, 2.007], p = <0.0001 (**I**) Thermal withdrawal latency was not significantly affected by intramedullary SP, n=8, M = (5.271; 4.691), t=0.9784, DF = 7, 95% CI [–0.8216, 1.981], p=0.3605. (**G, F**) students t-test *p<0.05, **p<0.01, ***p<0.001. n=5–7 males, 3 females/ group.

The online version of this article includes the following source data for figure 1:

**Source data 1.** Data for electrophysiolgical recordings and behavior from SP injection into the RVM.

## Intramedullary SP inhibits scratching

Since RVM ON cells are potentiated following injection of SP, we next tested if intramedullary micro-injection of SP affected itch and pain related behaviors. Mice were implanted with an intramedullary microinjection cannula dorsal to the RVM to allow microinjection of SP or vehicle (**Figure 1E**). The number of scratch bouts elicited by intradermal injection of histamine was significantly lower following intramedullary microinjection of SP compared to saline vehicle (**Figure 1F**). Scratching elicited by intradermal injection of chloroquine was also significantly reduced following intramedullary injection of SP compared to saline (**Figure 1G**).

There was a significant decrease in mechanical force to elicit a hindlimb withdrawal following intra-medullary SP compared to vehicle injection (**Figure 1H**), indicating mild mechanical sensitization. There was no significant effect of intramedullary SP injection on thermal hindpaw withdrawal latency (**Figure 1I**). We observed that both male and female mice showed a reduced hindlimb withdrawal, but that males had a significantly higher force for both saline and SP injection when compared with females (**Figure 3—figure supplement 3D**). Thus, activation of RVM Tacr1-expressing neurons, through intramedullary injection of SP, resulted in facilitation of mechanical nociceptive behavior and inhibition of pruritogen-evoked scratching behavior.

## Targeted expression of DREADDs in RVM Tacr1 expressing neurons

We next wanted to determine whether activation of RVM Tacr1 expressing neurons would modulate itch related behaviors. To selectively target Tacr1-expressing neurons in the RVM, we employed a chemogenetic approach. AAV-DIO-hM3dq-mCherry was injected into the RVM of Tacr1 cre mice,

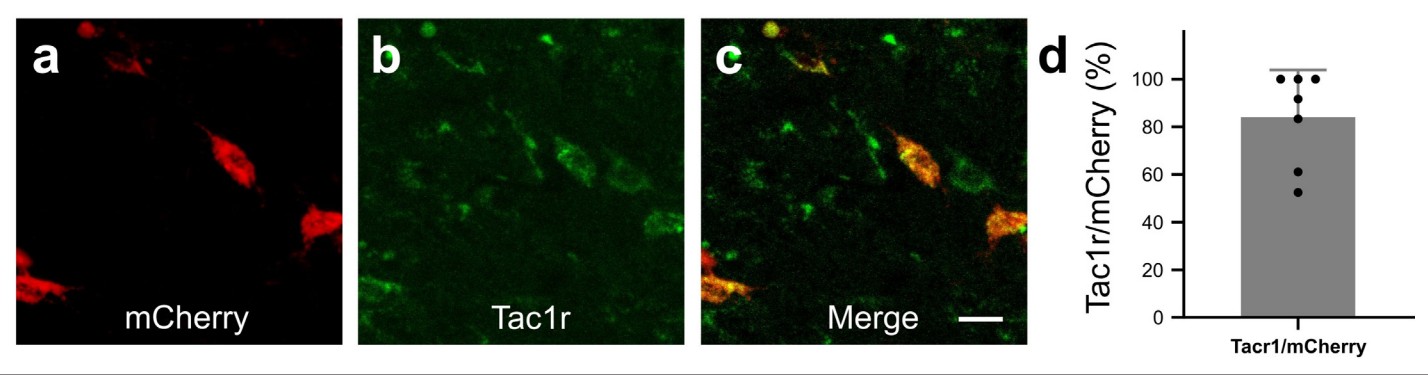

**Figure 2.** In Tacr1 cre $^{+/-}$ mice receiving AAV-DIO-hM3Dq-mCherry injections in the RVM, mCherry strongly colocalized with Tacr1 expression. (**A–C**): Images of RVM cells expressing mCherry (**A**), anti-Tacr1 antibody (**B**), and double-labeled cells (**C**). (**D**): 84% of mCherry expressing neurons co-express Tacr1. N=7. Scale bar (25 µm

The online version of this article includes the following source data for figure 2:

**Source data 1.** Data for co-expression of DREADDs to Tac1r.

resulting in selective expression of hM3Dq in Tacr1-expressing neurons (84% of RVM mCherry labeled neurons also expressed Tacr1 *Figure 2A–D*).

## Activation of RVM Tacr1 expressing neurons inhibits acute itch

To test the functional role of the RVM Tacr1-expressing population of neurons in itch modulation, we used DREADDs (hM3dq) to activate these neurons during pruritogen-evoked scratching behaviors (*Figure 3A*). Following viral injection, we observed clear expression of mCherry in the RVM (*Figure 3B*). Activation of DREADD-expressing RVM neurons in Tacr1 cre mice using clozapine significantly attenuated pruritogen-evoked scratching behavior. After clozapine administration (0.01 mg/kg, ip; *Gomez et al., 2017*), there was a significant reduction in scratch bouts elicited by intradermal histamine as compared to systemic administration of saline vehicle (*Figure 3C*). There was a significantly stronger inhibition of histamine-evoked scratching behavior in male compared to female mice (*Figure 3—figure supplement 3e*). Similarly, there was a significant reduction in chloroquine-evoked scratching (*Figure 3D*). In contrast, clozapine administration had no significant effect on the withdrawal threshold to mechanical von Frey stimuli (*Figure 3E*) or withdrawal latency to thermal stimulation (*Figure 3F*). Administration of clozapine in control vector (Tacr1-mCherry) mice had no significant effect on scratching behavior elicited by intradermal injection of histamine (*Figure 3C*) or chloroquine (*Figure 3D*). Likewise, clozapine administration had no significant effect on mechanical (*Figure 3E*) or thermal hindpaw withdrawals in vector controls (*Figure 3F*). Finally, neither a low (0.01 mg/kg) nor a higher dose (0.1 mg/kg) of clozapine had any significant effect on histamine- or chloroquine-evoked scratching behavior or on mechanically- or thermally-evoked paw withdrawals in wildtype mice (*Figure 3—figure supplement 2*).

Our results were independently confirmed using a separate line of Tacr1-CreER mice (*Huang et al., 2016*), which received intra-RVM microinjection of the excitatory DREADD AAV2-DIO-hM3dq-mCherry. Chemogenetic activation using CNO significantly reduced chloroquine evoked scratch bouts (*Figure 3—figure supplement 1A*) and spontaneous scratching behavior (*Figure 3—figure supplement 1B*), and also significantly reduced the von Frey mechanical withdrawal threshold (*Figure 3—figure supplement 1C*) with no effect on thermal withdrawal latency (*Figure 3—figure supplement 1D*). CNO had no effect in the control vector mice. Independent experiments in two lines of Tacr1 cre mice support the conclusion that activation of RVM Tacr1 expressing neurons inhibits pruritogen evoked scratching behavior.

## RVM Tacr1 expressing neurons inhibit chronic itch

Since activation of RVM Tacr1-expressing neurons inhibited acute pruritogen-evoked scratching behavior, we next wanted to determine whether chronic itch related behaviors would be affected. We used the imiquimod model of psoriasisiform dermatitis in Tacr1 cre mice receiving intra-RVM

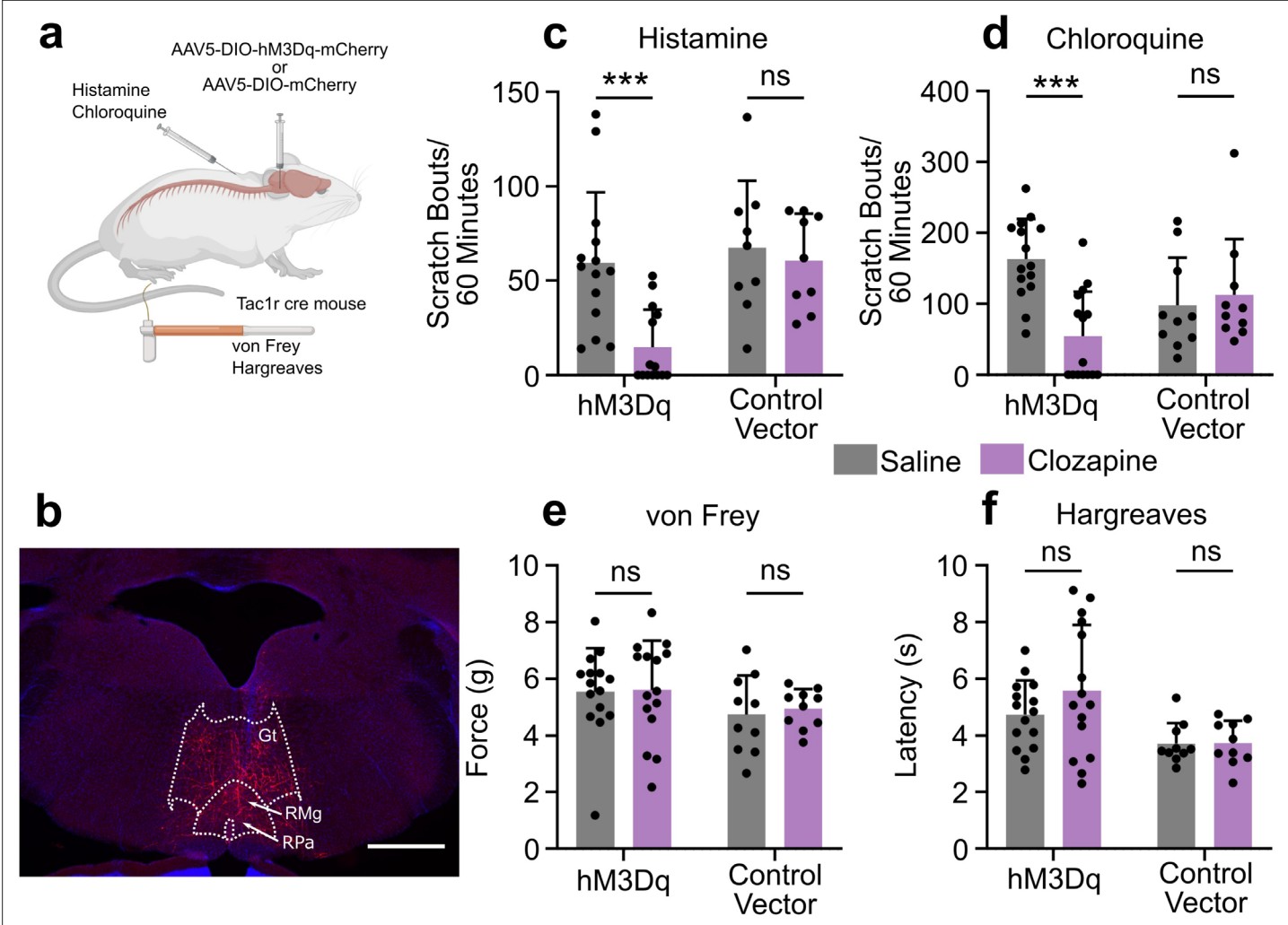

**Figure 3.** Chemogenetic activation of RVM Tacr1 expressing neurons inhibits itch related behavior. (**A**) AAV5-DIO-hM3Dq-mCherry or AAV5-DIO-mCherry was injected into the RVM of Tacr1 cre mice. (**B**) Expression of hM3Dq-mCherry was limited to the RVM. Gt: gigantocellularis; RMg: raphe magnus; RPa: raphe pallidus. Scale bar (1 mm). (**C**) Administration of clozapine caused a significant reduction in histamine-evoked scratching in hM3Dq expressing mice (7 males, 7 females, M = (59.32; 14.75), t=3.715, DF = 13, 95% CI [18.65, 70.49], p=0.0026), but not in control vector mice (six males, three females, M = (67.28; 60.61), t=0.6235, DF = 8, 95% CI [–24.11, 37.44], p=0.6235). (**D**) Administration of Clozapine caused a significant reduction in chloroquine-evoked scratching in hM3Dq mice (seven males, eight females, M = (163.2; 54.33), t=4.72, DF = 14, 95% CI [59.40, 158.3], p=0.0003) but not in control vector mice (six males, four females, M = (97.90; 112.9), t=0.4129, DF = 9, 95% CI [–23.45, 36.78], p=0.4129). (**E**) Clozapine administration did not significantly change mechanical withdrawal thresholds in hM3Dq (seven males, eight females, M = (5.549; 5.610), t=0.1309, DF = 14, 95% CI [–1.060, 0.9378], p=0.8977) or control vector mice (six males, four females, M = (4.742; 4.958), t=0.6342, DF = 9, 95% CI [–0.9867, 0.5546], p=0.5418). (**F**) Clozapine administration did not significantly change thermal withdrawal thresholds in hM3Dq (seven males, eight females, M = (4.735; 5.587), t=1.262, DF = 14, 95% CI [–2.300, 0.5961], p=0.2276) or control vector mice (six males, four females, M = (3.718; 3.737), t=0.0597, DF = 9, 95% CI [–0.7260, 0.6886], p=0.9537). (**C–F**) *p<0.05, **p<0.01, ***p<0.001, Students paired t-test.

The online version of this article includes the following source data and figure supplement(s) for figure 3:

**Source data 1.** Behavioral data of RVM Tac1r DREADDs activation with clozapine.

**Figure supplement 1.** Chemogenetic activation of RVM *Tacr1* expressing neurons inhibits itch-related behavior.

**Figure supplement 1—source data 1.** Behavioral data of RVM Tac1r DREADDs activation with CNO.

**Figure supplement 2.** Clozapine administration does not affect acute itch or pain behavior.

**Figure supplement 2—source data 1.** Data for Clozapine administration effects on behavior.

**Figure supplement 3.** Sex differences.

**Figure supplement 3—source data 1.** Data for behavioral sex differences.

injection of hM3Dq or control vector. Mice were treated daily with topical application of 5% imiquimod cream for 5 days. This resulted in a significant increase in spontaneous scratching on treatment day 3 (*Figure 4A*, dark blue) and significantly increased alloknesis scores on treatment days 1, 3, and 5 (*Figure 4D*, dark blue). Application of vehicle (Vanicream) had no effect on spontaneous scratching or alloknesis (*Follansbee et al., 2019*). Following administration of clozapine on day 5, there was a significant reduction in the number of spontaneous scratch bouts (*Figure 4A*, light blue). *Figure 4B* shows that scratch bouts in individual animals were significantly reduced after the administration of clozapine. Clozapine administered on days 3 and 5 also significantly reduced alloknesis scores (*Figure 4D*, light blue). *Figure 5E* shows significant reductions in alloknesis scores of individual animals after as compared to before clozapine. Mice that received intra-RVM microinjection of control vector also showed significant increases in spontaneous scratching (*Figure 4A*, dark green) and alloknesis scores (*Figure 4D*, dark green) following imiquimod treatment. Administration of clozapine did not significantly affect spontaneous scratching (*Figure 5C*) or alloknesis scores (*Figure 5F*) in these mice. These results show that activation of RVM Tacr1-expressing neurons reduces itch related behaviors in a model of chronic itch in addition to acute pruritogen-evoked scratch bouts.

## RVM ON cells express Tacr1

Previous reports had shown, in rats, that RVM ON cells were facilitated by local microinjection of SP (*Budai et al., 2007*; *Zhang and Hammond, 2009*) and ablation of RVM neurons which express Tacr1, was antihyperalgesic (*Khasabov et al., 2017a*), consistent with a pronociceptive role for RVM ON cells.

We used Tacr1 cre mice which were injected with an AAV encoding channelrhodopsin (AAV5:DIO-ChR2-eYFP). Four weeks later, the mice were anesthetized with sodium pentobarbital for single-unit recording with a microelectrode whose tip extended a few hundred micrometers beyond the tip of an optic fiber it was affixed to (*Figure 5A*). Injected mice exhibited robust expression of eYFP in the RVM (*Figure 5B*). ON and OFF cells were identified based on their response to a pinch stimulus. Once identified, blue light (473 nm, 0.25–5 mW) was applied and the cell was tested for entrainment to the light stimulus. *Figure 5C* shows an example of an ON cell in RVM that responded to pinch prior to the onset of EMG activity and was faithfully entrained to 5 hz light stimulation (*Figure 5D*). This neuron received a total of 37 light pulses at 2 hz and fired 32 action potentials within 20ms of the light onset (*Figure 5E*) with a calculated efficiency index of 0.86 (see below), and an average latency of 8.14ms. The latencies measured presently compare favorably with those reported previously for hippocampal neurons (*Zhang et al., 2013*). The latency, or on rate, of neuronal activation differs by cell type (*Herman et al., 2014*) and increases with diminishing light density (*Lin et al., 2009*). Out of 22 identified ON cells, 17 were entrained to the light stimulus, 1 was inhibited and 4 were not affected (*Figure 5F*). It is possible that our estimate is an under representation since viral transduction was not completely efficacious. At the end of the recording a lesion was produced which we histologically verified (*Figure 5J*). Since the majority of RVM ON cells were entrained to light stimulation, we conclude that the majority of ON cells likewise express Tacr1.

While our primary interest was RVM ON cells, we wanted to determine whether Tacr1 is also expressed in other cell types, such as RVM OFF cells. During our recordings we often found RVM OFF cells. Out of 14 identified OFF cells, none were directly activated by the light stimulus, with 7 showing a clear inhibitory response to the light stimulation while the other 7 were unaffected (*Figure 5F*). Often the degree of inhibition of the OFF cell increased proportionately to the stimulation frequency (*Figure 5G–I*). These results support the hypothesis that RVM ON cells provide an inhibitory input onto RVM OFF cells (*Fields and Heinricher, 1989*) and may underlie the RVM OFF cell mediated pause during noxious stimulation.

Occasionally, Neutral cells were identified by their response to optic stimulation (n=3), and each was strongly entrained (*Figure 5F*). Since Neutral cells are numerous, not affected by noxious stimulation and their contribution, if any, to descending modulation is unknown, they were not investigated further. We thus conclude that while most presently-recorded light-sensitive cells were ON cells, other cell types including Neutral cells also expresses Tacr1.

To validate our results from optotagging, we used several metrics to determine whether a light-sensitive neuron was truly entrained to the optic stimulus. We assumed that each optic stimulus was likely to directly excite a neuron expressing ChR2 and tested this in different ways. An efficiency index

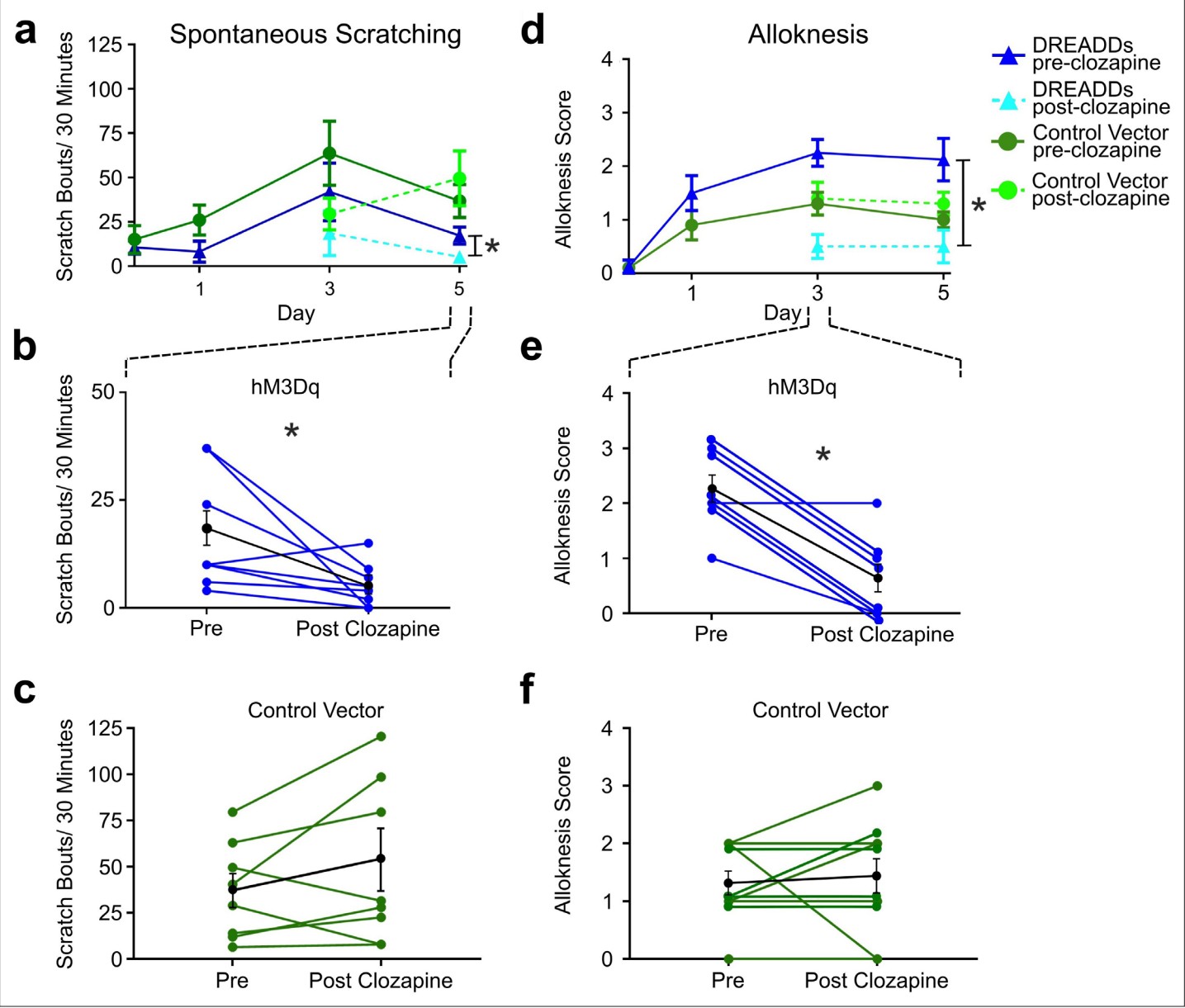

**Figure 4.** Chemogenetic activation of RVM Tacr1 neurons reduces spontaneous scratching and alloknesis in the imiquimod model of chronic psoriasisiform itch. (**A**) Application of imiquimod (1%, 0.05 g, Taro) once per day produced a significant increase in spontaneous scratching at day 3 for both DREADDs (dark blue, RM ANOVA, Dunnett's multiple comparisons Day0 vs Day 3, q=0.18511, p=0.0234) and control vector (dark green, RM ANOVA, Dunnett's multiple comparisons Day0 vs Day 3, q=2.549, p=0.0382) mice. Following clozapine, there was a significant reduction in spontaneous scratching on day 5 for the DREADDs mice (dashed light blue, Paired t test n=8, M = (17.25; 5.250), t=2.38, DF = 7, 95% CI [-23.92,–0.07597], p=0.0489) but no significant change in control vector mice (dashed light green, Paired t test n=10, M = (36.75; 49.56), t=1.343, DF = 9, 95% CI [–35.38, 9.752], p=0.2213). (**B**) Graph shows individual DREADDs animals' spontaneous scratch bouts pre- and post-clozapine. Following clozapine there was a significant reduction in scratching. Blue: individual counts; black: mean +/-SEM. Paired t test n=8, M = (17.25; 5.250), t=2.38, DF = 7, 95% CI [-23.92,–0.07597], p=0.0489 (**C**) Graph as in B for mice in vector control group, in which clozapine had no significant effect. Green: individual counts; black: mean +/-SEM. Paired t test n=10, M = (36.75; 49.56), t=1.343, DF = 9, 95% CI [–35.38, 9.752], p=0.2213 (**D**) Imiquimod induced significant increases in alloknesis scores on days 3 and 5 of treatment in DREADDs (dark blue, RM ANOVA, Dunnett's multiple comparisons, Day0 vs Day 3, q=9.379, p=0.<0.0001, Day0 vs Day5, q=4.00, p=0.0129) and vector control (dark green, RM ANOVA, Dunnett's multiple comparisons, Day0 vs Day 3, q=4.811, p=0.0025, Day0 vs Day5, q=5.014, p=0.0019) mice. Following clozapine administration on days 3 and 5 there were significant reductions in alloknesis scores for the DREADDs mice (dashed light blue, Paired t test n=8, M = (2.250; 0.6250) t=6.177, DF = 7, 95% CI [1.003, 2.247], p=0.0005) but not control vector groups (dashed light green, Paired t test n=10, M = (1.300; 1.400) t=0.3612, DF = 9, 95% CI [–0.7264, 0.5264], p=0.7263). (**E**) Clozapine resulted in a significant reduction in alloknesis scores (format as in B), Paired t test n=8, M = (2.250; 0.6250) t=6.177, DF = 7, 95% CI [1.003, 2.247], p=0.0005 (**F**) Clozapine had no

*Figure 4 continued on next page*

*Figure 4 continued*

effect on alloknesis scores in vector controls (format as in C). Paired t test n=10, M = (1.300; 1.400) t=0.3612, DF = 9, 95% CI [–0.7264, 0.5264], p=0.7263. Paired students t-test *p<0.05, **p<0.01, ***p<0.001. (**A–F**) n=4–5 males, 4–5 females/group.

The online version of this article includes the following source data for figure 4:

**Source data 1.** Data for RVM Tac1r activation in psoriasis model of chronic itch.

was calculated to determine whether a neuron was entrained to the optic stimulus, by counting the number of evoked action potentials divided by the number of optic stimuli (*Figure 5—figure supplement 1*). The 3 Neutral cells had the highest efficiency index (>1), which was due to occasional firing of 'doublet' action potentials in response to the optic stimulus (*Figure 5—figure supplement 1C, D*). ON cells classified as light sensitive also had a high efficiency index, which decreased with increasing optic stimulation frequency most likely due to desensitization of ChR2. We additionally assumed that the neuronal firing rate would reflect the number of optic stimuli. Indeed, the neuronal firing rate in light sensitive neurons increased with the stimulation frequency, while non-light sensitive neurons showed no change (*Figure 5—figure supplement 2*). In neurons which were inhibited by optic stimulation, there was a decrease in firing rate following light stimulation (*Figure 5—figure supplement 2*, OFF cells). Finally, we assumed that the light-evoked latency of neuronal action potentials would be consistent. For the population of light-sensitive ON cells, the average latency of responses to the optic stimuli delivered at 2 hz was 7.6 (+/-1.12) ms (*Figure 5—figure supplement 3*). Thus, these experiments strongly support the hypothesis that the majority of RVM ON cells express Tacr1.

## Discussion

Previous studies indicate a pronociceptive role for SP acting at Tacr1 expressing neurons in RVM (*Budai et al., 2007*), many of which have descending spinal projections (*Pinto et al., 2008*). A novel finding of the present study is that pharmacological and chemogenetic activation of Tacr1-expressing RVM neurons inhibited itch-related scratching behavior, with mild mechanical sensitization and no effect on thermal nociception. Our optotagging experiments provide evidence that the majority of ON cells express Tacr1. These results argue for a causal role of a subpopulation of Tacr1-expressing ON cells in the inhibition of itch-related scratching behavior, in contrast to their pronociceptive role.

### Role of SP in RVM in descending modulation of pain and itch

SP (*Ljungdahl et al., 1978*) and Tacr1 (*Saffroy et al., 2003*) are present within the RVM. SP acting in the RVM potentiated ON cells, induced pronociceptive behavioral effects and sensitized spinal wide dynamic range dorsal horn neurons (*Budai et al., 2007*; *Khasabov et al., 2017b*). SP appears not to be released in the RVM in the absence of injury, but is released under inflammatory conditions elicited by CFA or capsaicin to induce Tacr1-dependent hyperalgesia (*Brink et al., 2012*; *Hamity et al., 2010*; *Khasabov et al., 2017a*). The present study expands on this work in important ways. Firstly, we confirm that SP potentiates noxious pinch-evoked responses of mouse RVM ON cells. Secondly, our optotagging data provide evidence that the Tacr1-expressing neurons are ON cells. Importantly, our results support a role for these neurons in the inhibition of acute and chronic itch. Intra-RVM microinjection of SP or chemogenetic activation of Tacr1-expressing neurons in two different Tacr1 cre lines significantly decreased pruritogen-evoked as well as chronic itch related scratching behavior, while facilitating mechanical nociception in most experiments with no effect on thermal nociception. This latter observation contrasts with a previous report that intra-RVM microinjection of an SP agonist enhanced thermal nociceptive behavior in rats (*Khasabov et al., 2017b*) a discrepancy that might be attributed to a species difference.

### Descending modulation of itch

Optogenetic activation of GABAergic neurons in RVM facilitated mechanical, but not thermal nociception in mice (*François et al., 2017*), consistent with our data. The latter authors suggested that the GABAergic RVM neurons might represent ON cells, which descend to presynaptically inhibit mechanonociceptor input onto spinal inhibitory enkephalinergic interneurons that in turn contact the spinal

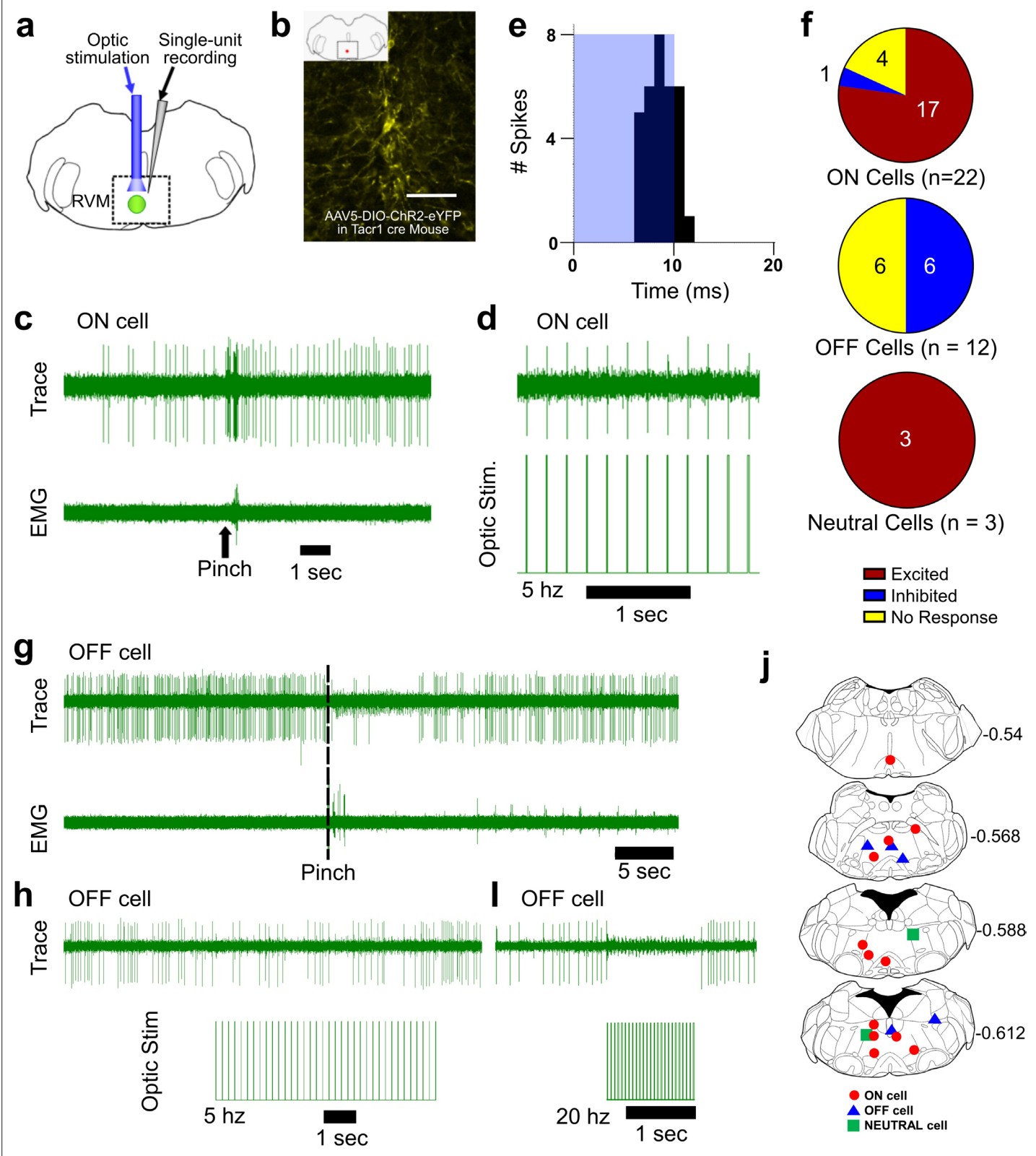

**Figure 5.** Optotagging of RVM Tacr1 neurons. (**A**) RVM cells were recorded with a microelectrode coupled to an optic fiber. (**B**) Injection of AAV-ChR2-eYFP in the RVM of Tacr1 cre+/- mice caused strong expression of eYFP. Scale bar (100 µm). (**C**) ON cells were identified based on their pinch-evoked response that preceded the hindlimb withdrawal as monitored by EMG in biceps femoris. (**D**) Cells identified as RVM ON cells were optically stimulated (5 mW, 472 nm). This neuron faithfully responded to each pulse in a 5 hz train. (**E**) Light entrainment was analyzed by creating peristimulus-time

*Figure 5 continued on next page*

*Figure 5 continued*

histograms (PSTH) of action potentials that occurred within a 20 ms window following the onset of each light pulse. This neuron responded consistently at a latency of approximately 8.14 ms with a calculated efficiency index of 0.86. (**F**) Distribution of RVM ON, OFF, and Neutral cells which were excited (red), inhibited (blue), or not affected (yellow) by optic stimulation. (**G**) Application of a pinch stimulus elicited a hindlimb withdrawal (dotted line) and a pause in firing that is typical of OFF cells. (**H**) There was an intermittent decrease in OFF cell firing during 5 hz optic stimulation and a (**I**) total cessation of firing during 20 hz optic stimulation. (**J**) Lesion sites from the optotagging recordings of RVM ON (red circles), OFF (blue triangles), and NEUTRAL cells (green squares). Numbers to right indicate Bregma coordinates.

The online version of this article includes the following source data and figure supplement(s) for figure 5:

**Figure supplement 1.** Efficiency index of classified RVM neurons.

**Figure supplement 1—source data 1.** Data for efficiency index analsyis of optotagging experiments.

**Figure supplement 2.** Firing rates of classified RVM neurons in response to optic stimulation.

**Figure supplement 2—source data 1.** Data for firing rate analsyis of optotagging experiments.

**Figure supplement 3.** Response latency of ON cells to optic stimulation.

**Figure supplement 3—source data 1.** Data for response latency of optotagging experiments.

mechanonociceptive pathway. Thus, activation of the descending GABAergic neurons would facilitate mechanical nociception via disinhibition.

Our study shows that RVM Tacr1 neurons represent a population of RVM ON cells, which when activated, reduce pruritogen-evoked scratching. A limitation of this study is that we were not able to directly demonstrate that inhibition of spinal pruriceptive transmission caused the reduction in scratching behavior. Previous studies have shown that 31% of functionally identified RVM ON cells (*Vanegas et al., 1984*), and 42.5% of Tacr1-expressing RVM neurons (*Pinto et al., 2008*), project to the spinal cord, supporting the descending modulation of spinal itch processing. Moreover, recent studies report that activation of spinal projection neurons originating in the locus coeruleus (*Koga et al., 2020*) or somatosensory cortex (*Wu et al., 2021*) suppress itch-related scratching behavior. Given this and the historical evidence of the spinal action of RVM ON cells, the most parsimonious explanation is that RVM Tacr1-expressing neurons with descending axons exert an inhibitory effect on spinal pruriceptive transmission to reduce scratching behavior. However, we cannot exclude the possibility that Tacr1-expressing RVM neurons lacking spinal projections exert an antipruritic effect via an unknown supraspinal action.

It was recently reported that activation of neurons in RVM that express the G-protein-coupled estrogen receptor (GPER) suppresses signs of acute and chronic itch (*Gao et al., 2021*). It is currently not known if GPER-expressing neurons co-express Tacr1.

Electrical stimulation of the PAG inhibited nocifensive behavior in rats (*Mayer et al., 1971*; *Reynolds, 1969*), and the PAG projects directly to the RVM (*Behbehani and Fields, 1979*). Activation of PAG GABAergic neurons, and inhibition of glutamatergic neurons, reduced scratching behavior under both acute and chronic itch conditions (*Samineni et al., 2019*) but facilitated nocifensive behaviors (*Samineni et al., 2017*). Activation of Tac1 (SP-expressing) neurons in the PAG was shown to facilitate pruritogen-evoked scratching via release of glutamate onto neurons in the RVM (*Gao et al., 2019*). In contrast, our results indicate that activation of RVM Tacr1 neurons inhibited pruritogen-evoked scratching behavior. This potentially represents a difference in the synaptic connections of PAG Tac1-expressing neurons with glutamate- and/or Tacr1-expressing RVM neurons. Previous studies have reported projections to RVM from SP-expressing neurons located in dorsolateral PAG, dorsal raphe nucleus, cuneiform nucleus, and lateral hypothalamus (*Chen et al., 2013*; *Holden and Pizzi, 2008*; *Yin et al., 2014*). It is thus possible that Tacr1-expressing RVM neurons that inhibit scratching are activated by different SP inputs than the RVM neurons activated by PAG TAc1-expressing neurons that facilitate scratching.

## Modality specific role of RVM ON cells

For decades, the role of RVM ON cells has been considered faciliatory for spinal nociceptive transmission. Our results suggest that the role of RVM ON cells is modality specific, with RVM ON cells facilitating spinal nociceptive while inhibiting pruriceptive transmission. Pruritogens and algogens similarly excited RVM ON and inhibited OFF cells, implying that the opposing modulatory effects of ON cells on spinal nociceptive vs. pruriceptive transmission likely occurs via separate spinal circuits. RVM ON

cells have a GABA-mediated inhibitory connection to spinal enkephalinergic neurons (*François et al., 2017*), raising the possibility that spinal enkephalinergic tone accounts for the opposing effects on spinal nociceptive and pruriceptive transmission. Since pain suppresses itch, we cannot exclude the possibility that activation of spinal pronociceptive circuitry inhibits itch transmission.

### Treatment of chronic itch with Tacr1 antagonists

The Tacr1 antagonists aprepitant and serlopitant were recently shown to be partially effective in reducing chronic itch of various etiologies. Tacr1 antagonists might act at Tacr1 that is expressed by spinothalamic and spinoparabrachial neurons (*Yang et al., 2021* & Todd, 2010) to reduce pruriceptive transmission. Indeed, ablation of Tacr1-expressing spinal and medullary neurons reduces pruritogen-evoked scratching behavior (*Carstens et al., 2010*). The present results suggest that Tacr1 antagonists also have an opposite effect to block Tacr1-mediated descending inhibition of itch. Systemic administration of Tacr1 antagonists thus appear to have offsetting supraspinal and spinal effects, potentially explaining the limited efficacy of Tacr1 antagonists for chronic itch.

## Materials and methods

**Key resources table**

| Reagent type (species) or resource | Designation | Source or reference | Identifiers | Additional information |
|---|---|---|---|---|
| Strain, strain background (*M. musculus*) | Tacr1 cre | Dong Lab | | |
| Strain, strain background (*M. musculus*) | Tacr1 creER | Dong Lab (*Huang et al., 2016*) | | |
| Strain, strain background (AAV) | aav5-hSyn-DIO-hM3Dq-mCherry | Addgene | 44361-AAV5 | |
| Strain, strain background (AAV) | aav5-hSyn-DIO-mCherry | Addgene | 50459-AAV5 | |
| Strain, strain background (AAV) | aav5-Ef1a-hChR2-mCherry | Addgene | 20297-AAV5 | |
| Antibody | Mouse monoclonal: anti-Tacr1 conjugated 488 | SCBT | sc-365091 AF488 | (1:50) |
| Chemical compound, drug | Substance P (SP) | Tocris | 1156/5 | |
| Chemical compound, drug | Histamine Dihydrochloride (HA) | Sigma-Aldrich | H7250 | |
| Chemical compound, drug | Chloroquine Diphosphate (CLQ) | Sigma-Aldrich | C6628 | |
| Chemical compound, drug | Pentobarbital Sodium | Sigma-Aldrich | P3761 | |
| Chemical compound, drug | Buprenorphine Hydrochloride | Amerisoucebergen | NDC42023-179-05 | |
| Software, algorithm | 5 Prism | Graphpad | | |

### Animals

Experiments were performed using wild type mice (c57BL/6 j, Jackson Labs, Bar Harbor ME), Tacr1 cre$^{+/-}$ (courtesy of Dr. X. Dong, Johns Hopkins University), and Tacr1 creER $^{+/-}$mice (*Huang et al., 2016*) of both sexes, 8–10 weeks of age, all on a C57Bl/6 background. Mice were given free access to food and water and housed under standard laboratory conditions. Mice were housed in-lab and at the animal housing facility, with a natural light cycle and 12 hr light dark cycle, respectively. Mice were allowed to habituate for at least 3 days following transfer from the animal facility to lab housing before use in behavioral experiments. Mice were cohoused with between 1 and 4 mice/cage. All procedures were approved by the University of California, Davis and University of Pittsburgh Animal Care and Use Committees and followed the ARRIVE guidelines to the extent possible.

### Pharmacologic agents

Clozapine was dissolved in saline and administered intraperitoneally (ip) at concentrations of 0.01 and 0.1 mg/kg. Clozapine-N-oxide (CNO;Tocris, Bristol UK) was dissolved in phosphate-buffered saline and administered ip (5 mg/kg). In clozapine and CNO treated mice, experiments were conducted 30 min following their administration. Histamine HCl (Sigma, St. Louis MO; 0.5% in 10 µl) or chloroquine

diphosphate salt (Sigma; 1% in 10 μl) were dissolved in physiological saline and administered intradermally via a 30 g needle in the nape of the neck. Imiquimod cream (5%; Aldara, 50 mg; 3 M Health Care Limited, UK) was administered topically once per day to shaved skin on the rostral back for 5 days.

## Stereotaxic injections and cannula implantation

Animals were anesthetized with 2% isoflurane and placed in a stereotaxic head frame. A burr hole was made in the calvarium and a Hamilton microsyringe loaded with virus was stereotaxically placed such that the tip was at the injection site in RVM (RC: –5.5 to –5.8 mm, ML: 0.0, DV: –4.2 to –6 mm). Virus (0.25–1 μl) containing either AAV5: hSyn-DIO-hM3Dq-mCherry (excitatory DREADD; Addgene, Watertown MA), AAV5: hSyn-DIO-mCherry (control vector, Addgene), or AAV5: hSyn-DIO-ChR2-eYFP (Addgene) was injected into the RVM. Virus was infused at an approximate rate of 100 nL/min. The injection needle was left in place for an additional 15 min post-injection and then slowly withdrawn. The incision was closed using Vetbond and animals were given ketofen (10 mg/kg ip) or buprenorphine (0.05 mg/kg ip) and allowed to recover on a heating pad. A 4-week recovery period ensued prior to experimentation.

For intracranial drug injections, an injection cannula (Plastics One, Roanoke VA) was implanted stereotaxically with the tip targeted 1–2 mm above the RVM. For optogenetic stimulation, following microinjection of AAV5: hSyn-DIO-ChR2-eYFP into RVM, an optic fiber (Doric Lenses, Quebec Canada) was stereotaxically implanted at the target site. Dental cement was used to affix the injection cannula or optic fiber to the skull. Post-implantation mice received buprenorphine (0.05 mg/kg ip). Mice recovered for at least 4 weeks prior to experimentation.

## Intracranial drug microinjection

Mice were habituated for 15 min prior to intracranial drug injection in clear cylindrical glass enclosures and videotaped from above. Saline (0.5 μl) or Substance P (SP, 0.5 μl, 10 nmol, Tocris, Minneapolis MN) was microinjected. Fifteen minutes post-injection mice were tested for itch or pain behaviors as described below. Both male and female mice were used.

## Chemogenetics

Thirty minutes after ip administration of saline, clozapine or CNO, behavioral testing commenced. We verified the efficacy of clozapine by observing an increase in cfos staining in the RVM of DREADDs versus control vector mice as well as saline versus clozapine (data not shown).

## Behavior

Scratching behavior: Scratch bouts were defined as back-and-forth hindpaw movements directed to the rostral back, followed by biting the toes and/or placement of the hindpaw on the floor. Scratch bouts elicited by histamine, chloroquine or saline vehicle were videotaped and counts made by at least two blinded observers. Nociceptive behavioral assays (von Frey; Hargreaves) were conducted by investigators blinded as to treatment. Successive behavioral tests were conducted in pseudorandom order and spaced 1 week apart.

Von Frey: Mechanical sensitivity was measured in two ways. Mice stood on a mesh floor allowing access to the plantar surface from below. Using the Chaplan up-down method (*Chaplan et al., 1994*), calibrated von Frey filaments (North Coast Medical Inc) were applied to the plantar surface. Paw lifting, shaking, and licking were scored as positive responses. Averaged responses were obtained from each hindpaw, with 3 min between trials on opposite paws, and 5 min between trials on the same paw. Alternatively, the force at the moment of hindpaw withdrawal was measured using an electronic von Frey device (2390; IITC, Woodland Hills CA). Measurements were again taken from each hindpaw with 3 and 5 min between trials on the same or opposite paw.

Hargreaves: Animals were acclimated on a glass plate held at 30 °C (Model 390 Series 8, IITC Life Science Inc). A radiant heat source was applied to the hindpaw and latency to paw withdrawal was recorded (*Hargreaves et al., 1988*). Two trials were conducted on each paw, with at least 5 min between tests of opposite paws and at least 10 min between tests of the same paw. To avoid tissue damage, a cut off latency of 20 s was set. Values from both paws were averaged.

## In vivo single-unit recording

Adult mice (6 weeks) mice were anesthetized with pentobarbital sodium (60 mg/kg, ip). The head was secured in a stereotaxic frame and an opening was made in the occipital bone. The animal's body

temperature was maintained with a heating pad and external heating source. Teflon coated silver wires were inserted into the biceps femoris to record electromyographic (EMG) activity. A single-unit recording microelectrode (10 MOhm, Frederick Haer Inc, Bowdoin ME) was coupled to a 33-gauge injection cannula such that the tip of the recording microelectrode extended several hundred µm beyond the tip of the injection cannula and was inserted into the RVM. ON cells were identified by a hindpaw pinch-evoked increase in firing that preceded the hindlimb withdraw reflex measured as EMG activity from fine wires inserted in the biceps femoris. Once an ON cell was identified, either saline or SP (0.5 µl, 10 nmol) was microinjected and changes in firing rate to repeated consistent pinch stimuli were recorded. Responses to pinch stimuli were normalized to the initial response for each unit. Electrode voltages were amplified and digitized (CED 1401, CED, Cambridge UK) and analyzed with Spike2 (CED). At the end of the recording, a lesion was produced at the last recording site by passing direct current through the microelectrode, the brain was harvested postmortem and postfixed in 10% formalin.

## Optotagging

At least 4 weeks following injection of AAV5: hSyn-DIO-ChR2-eYFP into RVM, single-unit recordings were made from RVM as described above using a Tungsten microelectrode attached to an optic fiber such that the microelectrode tip extended a few hundred microns beyond the optic fiber. In most experiments ON cells were functionally characterized as described above, and OFF cells were characterized by a pinch-evoked pause in ongoing activity that preceded the withdrawal reflex. ON and OFF cells were then tested for entrainment to optic stimulation at 473 nm wavelength and 0.25 mW - 5 mW light output (5–20 Hz, 10 ms pulse duration) from a laser (Laserglow R471003GX). In some experiments, optic stimulation was used as to isolate light-sensitive neurons, which were then identified as ON, OFF or Neutral based on their response to pinch. Efficiency indices were calculated as the number of action potentials firing within a 20 ms window following the onset of the optic stimulus divided by the total number of optic stimulus pulses, and were used to determine whether a neuron was entrained to the optic stimulus. Both male and female mice were used for all electrophysiological experiments.

## Imiquimod treatment

Imiquimod cream was applied topically to the shaved area on the rostral back once per day for 5 consecutive days. Treatment groups consisted of age-matched male and female NK-1-cre mice that had received intra-RVM injection of AAV5: DIO-hM3Dq-mCherry or the control vector AAV5: hSyn-DIO-mCherry. Imiquimod treatment induced signs of skin pathology including skin scaling and erythema. As measures of chronic itch, we assessed spontaneous scratching, and alloknesis, 23 hr following imiquimod treatment. Mice were videotaped and tested between 10 AM and 5 PM, with each individual mouse tested at the same time each day. The mice were habituated to glass cylinders for 3 successive days prior to recording. Animals were videotaped from above for 30 min. Behavioral videos were analyzed by two blinded observers. Only discrete bouts of spontaneous hindlimb scratches directed towards the application site were counted, as described previously (*Akiyama et al., 2016*) and summed over the 30 min period. Alloknesis was assessed as previously described (*Akiyama et al., 2012*). The mouse was placed in an enclosed area and a 0.07 g von Frey monofilament was applied to the perimeter of the imiquimod application area five consecutive times. The alloknesis score consisted of the number of immediately occurring hindlimb scratch bouts directed to the stimulus site.

## Immunofluorescence

Four weeks after intra-RVM injection of AAV5:hSyn-DIO-hM3Dq-mCherry or the control vector (AAV5:hSyn-DIO-mCherry) in Tacr1 cre mice, clozapine (0.01 mg/kg, i.p.) was injected, followed 1 hr later by perfusion (4% paraformaldehyde), harvesting of brains and post-fixation overnight in 4% paraformaldehyde. Brains were sectioned (20 µm) on a freezing microtome and stained as free floating sections. DREADDs-expressing neurons were counterstained for Tacr1 expression with a conjugated Tacr1 antibody (D-11, sc-365091, Santa Cruz Biotechnology, Dallas TX) at 1:50 overnight at 4 °C. mCherry expressing neurons were identified in the RVM and then observed for Tacr1 expression. All slides were mounted with vectashield and imaged with confocal microscopy. Staining intensity was

measured relative to the red fluorescence (from DREADDs) and was quantified using FIJI (*Schindelin et al., 2012*).

## Statistical analysis

All statistical analyses were performed using GraphPad Prism. Values are presented as mean +/-SEM. Statistical significance was assessed using students t-test or a two-way, repeated measures ANOVA with Bonferroni's correction, unless otherwise specified. Significance was indicated by $p < 0.05$. Sample sizes were based on pilot data and are similar to those typically used in the field.

For analysis of effects of intracranial microinjections, a paired students t-test was used to compare the effects of intracranial injection of saline versus SP on behavioral measures. Two-way ANOVA determined a lack of significant sex x SP microinjection interaction, so the data from both sexes were pooled. For chemogenetic experiments, a paired t-test was used to compare scratch counts and nociceptive measures following vehicle vs. clozapine or CNO injection, or between vector controls and DREADDs mice following clozapine or CNO. For optogenetic experiments a paired t-test was similarly used to compare behavioral measures during and in the absence of optic stimulation.

For experiments with imiquimod, paired students t-tests compared the effects of ip injection of saline versus clozapine on spontaneous scratch bouts and alloknesis scores. A two-way ANOVA revealed no interaction for sex x clozapine administration, so the data were pooled for further analysis.

For single-unit recordings, a two-way repeated measures ANOVA with bonferroni post-hoc test was used to compare the neuronal responses to pinch following intra-RVM injection of saline or SP for 60 min post-injection.

For immunohistochemical staining for c-fos, students t-test compared the staining intensity of c-fos and the proportion of c-fos-positive neurons in Tacr1-expressing neurons following administration of clozapine, in mice receiving DREADDs or control vector injections.

## Acknowledgements

We thank Dr. Dong (Johns Hopkins University) for the generous donation of Tacr1-cre mice.

## Additional information

### Funding

| Funder | Grant reference number | Author |
| --- | --- | --- |
| National Institutes of Health | AR076434 | Earl Carstens |
| National Institutes of Health | AR057194 | Earl Carstens |
| National Institutes of Health | AR063772 | Sarah E Ross |
| National Institutes of Health | NS096705 | Sarah E Ross |
| National Institutes of Health | F31NS113371 | Eileen Nguyen |
| National Institutes of Health | T32GM008208 | Eileen Nguyen |

The funders had no role in study design, data collection and interpretation, or the decision to submit the work for publication.

### Author contributions

Taylor Follansbee, Conceptualization, Resources, Data curation, Software, Formal analysis, Supervision, Validation, Investigation, Visualization, Methodology, Writing – original draft, Project administration, Writing – review and editing; Dan Domocos, Formal analysis, Funding acquisition, Investigation; Eileen Nguyen, Amanda Nguyen, Lauren Velasquez, Data curation, Investigation; Aristea Bountouvas,

Investigation; Mirela Iodi Carstens, Resources, Supervision, Investigation, Methodology, Project administration; Keiko Takanami, Data curation, Formal analysis, Investigation; Sarah E Ross, Supervision, Writing – review and editing; Earl Carstens, Conceptualization, Data curation, Formal analysis, Supervision, Funding acquisition, Methodology, Writing – original draft, Writing – review and editing

## Author ORCIDs

Taylor Follansbee  http://orcid.org/0000-0002-4470-9324
Sarah E Ross  http://orcid.org/0000-0003-2593-3133
Earl Carstens  http://orcid.org/0000-0003-4696-8116

## Ethics

This study was performed in accordance with our approved IACUC protocol (21316) at the University of California, Davis.

## Decision letter and Author response

Decision letter https://doi.org/10.7554/eLife.69626.sa1
Author response https://doi.org/10.7554/eLife.69626.sa2

## Additional files

### Supplementary files
• Transparent reporting form

### Data availability
All data generated or analyzed during this study are included in the manuscript and supporting files.

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
