## [Editor Report]

This manuscript uses several complementary strategies to investigate and manipulate the activity of neurokinin 1 receptor-expressing neurons in the rostroventral medulla of mice in the context of itch sensation. The study delivers important insights into supraspinal itch processing and descending modulation of itch signals in the spinal cord.

---

## [Decision Letter]

**Decision letter after peer review:**

Thank you for submitting your article "Descending inhibition of itch by neurokinin 1 receptor (Tacr1) -expressing ON cells in the rostral ventromedial medulla" for consideration by *eLife*. Your article has been reviewed by 3 peer reviewers, and the evaluation has been overseen by a Reviewing Editor and Kate Wassum as the Senior Editor. The following individuals involved in review of your submission have agreed to reveal their identity: Mark Hoon (Reviewer #2); Vijay K Samineni (Reviewer #3).

Essential revisions:

1. Perform further experiments to show more convincingly that descending inhibition of itch signal in the spinal cord is the mechanism the tacr1 rvm 'on' neurons are using to reduce scratching. Consider direct functional evidence for descending modulation, as opposed to indirect spinal consequences of local modulation in the RVM, and report the proportion of the tacr1 rvm neurons that send descending projections to the spinal itch circuitry.

2. Include essential controls in experiments involving DREADDs

3. Address the caveat of heterogeneity of tacr1 cells in association with the claim of their identity as ON cells; accordingly either provide additional evidence or temper down claims.

4. Undertake changes in text and figures recommended by reviewers under the 'recommendations' section of individual reviews, as appropriate

5. If you have not already done so, please ensure your manuscript complies with the *eLife* policies for statistical reporting: https://reviewer.elifesciences.org/author-guide/full Report exact p-values wherever possible alongside the summary statistics, degrees of freedom, and 95% confidence intervals. These should be reported for all key questions and not only when the p-value is less than 0.05.

6. Please also include a key resource table.

7. Please include housing details, including number of animals/cage (e.g., single v. pair v. group housed), light-dark cycle, cycle at test etc.

*Reviewer #1 (Recommendations for the authors):*

Decades of penetrating investigation of nociception at the level of the brainstem has revealed a complex circuitry including ON and OFF cells that activate or deactivate, respectively, in response to noxious input. This manuscript begins to unwind the relationship of these cells and circuits to the signaling of itch. The main conclusion, that a population of Tacr1 expressing neurons in the rostroventral medulla are "ON" cells and that activation of these cells suppresses scratching is supported by anatomical, behavioral, and functional data. The strengths of these conclusions rest on the use of several approaches to manipulate activity (optogenetic, chemogenetic, and pharmacological) and from single unit recordings in anesthetized animals, which provide a high level of internal consistency. The use of appropriate blinding, detailed analysis of sex differences, and several experiments offering detail on the precision of optogenetic activation are other methodological strengths. While the data are strong, the interpretations at times stray into areas of less certainty, and a singular vision is presented where alternate possibilities still exist.

One alternate possibility is that the activation of Tacr1 RVM neurons with substance P enhances nociception, and it is this enhanced pain that limits the scratching behavior, rather than a decrease in the itch sensation per se. This is supported by the lowered mechanical withdrawal threshold after DREADD activation of TacR1 neurons. Perhaps scratching becomes too painful, or perhaps the effectiveness of scratching is increased, and therefore fewer scratches are required for the same itch suppressing effect. Although Liu 2019 is cited in support (p. 20) of the authors' conclusions, there is not data in the present manuscript showing that the Tacr1 RVM ON cells under study directly modulate the spinal itch circuitry, or the GRPR cells.

The experiments in figure 2 are missing a control – DREADDs may have some constitutive activity, and therefore it is important to show with and without the DREADD activator CNO or clozapine (the authors use both interchangeably for some reason).

There is also the nagging problem that somehow motor systems are compromised by the activation of RVM neurons (the spinal cord appears to be fully labeled with mcherry Figure 4f, including the ventral horn). Although the contrasting findings between less scratching and more von Frey withdrawal suggest an intact system at the most basic level, more convincing would be showing enhanced scratching activity via inhibition of the Tacr1 RVM ON cells (a positive behavioral output). This would both put aside motor concerns and strengthen the functional role of these cells. Since according to the recordings in Figure 6c, ON these on cells have ongoing activity prior to the pinch stimulus, it should be possible to inhibit this tonic activity.

Figure 7 should include more of the findings from the results. No OFF cells are represented, lack of involvement of 5HT not shown.

Finally, the text generally needs more methodological details to understand the experiments. I realize this is a trend with some journals to put the methods at the end, which is fine, but additional basic information must be presented to understand and interpret the experiments. Specific instances are listed below.

Line 78+ state briefly what prep is being used (anesthetized, in vivo).

Indicate volume and concentration of drugs (ie substance P, hist, cq). This could even be in figure legend.

Figure 1 (also Figure 6) Please confirm accuracy of the Bregma coordinates.

Line 86: “enhancement of responses lasted 1 hr” please show.

Figure 1 e: it looks like blue light is shining in this figure. Please use a different style for clarity.

Figure 2 – Are the figures a and e mixed up?

Figure 2 – I couldn't understand the value of including i-k, seems duplicative.

Figure 2 – legend “mean number of c-fos”, but figure shows percentage, correct; also see line 119-121; again, this needs a no CNO control.

Figure 3B would be more convincing for the authors’ statement that “expression of hM3Dq-mcherry was limited to the RVM” if the area outside the RVM was included in the figure.

Data between Figure 4 and Figure 3 are a bit different re: von Frey. Please explain the added value of the tac1rcre-er mice versus the previous mouse line and offer some insight into this discrepancy.

Figure 5 – this figure was awkward (particularly a and b). What is meant by the bars and stars on the side? The shades of green are hard to distinguish. The legend states “imiquimod…produced a significant increase in spontaneous scratching” – but I could not identify the statistical test used. There is an indication that something is “red”, but I see no red in the figure. Why is clozapine used instead of CNO? Why are imiquimod-evoked scratches “spontaneous”?

Line 187 – “tip extended a few millimeters” This seems too far. I’m not sure it would work properly. The methods says micrometers.

Line 189 (and elsewhere), light used is given as less than 5mW. This is vague, and could be 0mW.

Line 191 – “faithfully entrained 10Hz (6d)” but only 5 hz is shown.

Line 219 – “doublet” action potentials. I was expecting a picture.

Line 227 – Figure 4 (should be supplemental).

Line 234 – reword; it sounds like the actual force of the limb withdrawal was greater.

Line 246 – Please define what is the fev-cre line upon first introduction.

Line 305 – reference to Figure 4, but I think maybe Figure 2 is meant?

*Reviewer #2 (Recommendations for the authors):*

Strengths: The results from this study broadly indicate that RVM TacR1-neurons can alter itch responses and are mainly ON-cells which are new findings that define important elements in the itch sensory system.

Weaknesses: neurochemical experiments suggest that TacR1 is expressed in a heterogeneous population of RVM neurons. Further, the results from the use of two different mouse lines where TacR1-neurons are chemogenetically manipulated, differ from each other. These two issues make it difficult to completely unambiguously interpret results from chemogenetic experiments.

Overall, the data in this manuscript is interesting and show important new findings. However, several of the interpretations made in the manuscript go beyond the results described and a revision of the manuscript is needed to reasonably reflect the presented results. There are also other questions about the Results and errors in the way that data is presented. First, the TacR1-neurons recorded from and manipulated are not directly shown to be descending. This is inferred from other studies and spinal cord staining but, what proportion of them are descending and could some of the effects reported be through indirect pathways? Further, the study does not directly establish that the effects seen occur in the spinal cord. For instance, some of the TacR1 neurons may be interneurons and some of the effects of activating TacR1 neurons may occur in other brain regions. Second, another concern is the incomplete characterization of the TacR1-cre mouse lines; the second TacR1-creERT2 mouse line does not seem to have been neurochemically characterized and the characterization of the one TacR1-cre line is not thorough (see comments below). Third, the parts of the manuscripts on 5HT signaling are not well connected with the other parts of the paper and need to be integrated or removed. Fourth, although it is not unreasonable to suggest that the OFF-cells are downstream of ON-cells, the manuscript does not provide direct proof of this; TTX, GABA inhibition experiments and measurements of latency responses relative to optogenetics are probably required to demonstrate this.

I realize that all these requests would likely require considerable additional experimentation, and this may go beyond *eLife* guidelines. In the absence of performing new extra studies, the title, abstract and discussion will all need revisions to tone down conclusions to better reflect the data presented.

Below is an itemized list of concerns:

a) The data in figure1c is a bar graph, scatter plots should be used to allow a better assessment of the data.

b) Figure 1h, are the values on the y-axis correct? (8-2g, presumably 0.8g to 0.2g). This increase in sensitivity would be better described as a mild mechanical sensitization, similarly data in figure 3 should be described as mild-mechanical sensitization.

c) Figure 2, panels a and e have been switched. The staining for mCherry is different for each panel, what is the reason for this? – are they the same region or is A in the reticular region of the RVM and other images taken in the midline? i.e. were images taken from the same area of the RVM and if not, how was data normalized? Since the RVM encompasses several separate brain nuclei, the authors should detail these, for instance; raphe magnus, raphe pallidus, PGRNL, and MARN. In addition, in the electrophysiology experiments described in figure 6, ON-cells and OFF-cells were probed in many areas of the general medulla – was the characterization of the lines used also similarly widely distributed? Images in figure 2 (or a new supplement) should be of whole coronal sections of the medulla in order to see the locations of all the TacR1-neurons in the medulla.

d) Images in figure 2ab and c indicate 4 double positive and 4 TacR1 alone neurons, but the graph reports there is an 80% overlap. A move representative image should be presented.

e) Figure 2d, h, and l should be presented as scatter plots.

f) Figure m, n, o same comment as d), an image showing more neurons would better illustrate the data presented in panel p.

g) Figure 2m shows endogenous TacR1 expression and does not report cells where cre mediated excision occurs. Therefore, this does not prove that cells chemogenetically manipulation or recorded from are GABAergic. The ISH should be on tissues from TacR1-cre mice infected with AAV-cherry and if different serotypes of AAV were used in different experiments this should be noted as a limitation of the study.

h) All numbers of counted neurons for figure 2 need to be reported.

i) Were similar neurochemical characterization studies performed on the second TacR1-creERT2 line? If they were not, then the authors should acknowledge that the differences in results between these lines may be because different populations of neurons might be activated.

j) Line 115, sentence needs rewording.

k) Line 129, this sentence repeats the previous one and is less accurate giving a false impression the presented data shows that all Tacr1-neurons are inhibitory. Similar inaccuracies are repeated throughout the manuscript.

l) Are the TacR1-neurons the same as those described in Francois et al. 2017?

m) Different serotypes of AAV are used in Figure 3 and 4, a comment on this should be added as this may additionally contribute to the differences in effects of chemogenetic manipulation between these experiments.

n) 50ug of chloroquine produces different numbers of itch bouts in controls in figure 3 and 4, what's the reason?

o) Given that these experiments are suggested to confirm each other, is there a reason why in figure 3 von Frey thresholds are not affected but in figure 4 they are? This difference needs to be acknowledged in interpreting these results.

p) Figure 4e, the staining for mCherry seems to extend beyond the cell bodies of neurons in the raphe magnus (dorsal and lateral), are these dendrites and could the AAV retrogradely infect, in the RVM, terminals of neurons in other brain regions? A larger image would help resolve whether the staining is from dendrites.

q) Figure 4f, the contrast used for this image makes it hard to see any staining. Is there bilateral staining and which lamina are these terminals projecting to and are projections the same in all spinal cord segments?

r) The electrophysiology of optogenetic responses of ON and OFF-cells- optotagging. What were the waveforms and latency of the proposed directly activated versus indirectly activated neurons and were there any indications there is more than one class of ON-cells? Was supervised/unsupervised clustering used to distinguish cell-types? Even though the evoked response reliability of ON-cells is high, the latency of these neurons in slow compared to other optotagging studies reported. What is the response reliability of OFF-cells is the distribution of latency for ON-cells suggest more than one class? And what's the average latency of response of OFF-cells? Were experiments with NBQX and bicuculine performed to show direct activation?

s) Figure 6b shows a small area of where recordings were made (versus panel j). A larger view would be good.

t) Were studies performed to show ON-cells are synaptically connected to OFF-cells? I.e. for OFF-cells could responses be blocked with TTX, or since the TacR1-cells are suggested to be mainly GABAergic are these responses blocked with bicuculline? Without these results, it is possible that TacR1 is expressed in both ON and OFF-cells.

u) The section on sex differences is odd. It needs to be integrated with the other parts of the paper.

v) Section of 5HT RVM cells does not connect with other parts of the paper and basic control data are not presented for these studies. I would recommend this section be removed as it does not add to the main theme of the paper and the data is too preliminary.

*Reviewer #3 (Recommendations for the authors):*

The manuscript looks at contributory role of a RVM Tacr1-expressing neurons in the acute and chronic pruritis. Authors demonstrate that RVM ON and neutral cells express Tacr1 and send projections to the spinal cord, implicating possible role in descending inhibition of spinal pruriceptive transmission. Overall, the study used well established and complementary methods to support their findings. This work fills significant gaps in the descending itch literature. Largely I think the paper is well presented, data supports the conclusions of the study. There are some concerns regarding the figures. The manuscript could improve if authors address these concerns.

1. Authors show that in Figure 1, SP injections into the RVM significantly reduced evoked itch and von Frey behaviors. It is confusing as the data is presented, authors say they are using Paired student test. But in the methods, it is not clear if they use same mice for saline and SP injections. Paired T test would be appropriate if pre and post were conducted on same mice.

2. SP microinjections did not always decrease evoked itch behaviors. Any differences observed in the location of injections that explains this variance in the data?

3. Figure 2a and 2e seemed to be switched. Authors should address this.

4. cFos is a nuclear protein. In the images authors show, it seems like they are just capturing background autofluorescence instead of cFos signal.

5. Authors claim Tacr1 neurons are Vgat+ve. Could authors comment on if they have looked to see any Tacr1 neurons are Vglut positive. It is possible because it has not been tested there could be neurons Tacr1+ve neurons that could be Vglut+ve.

6. In Figure 3, authors seem to use same panels for histamine and chloroquine evoked scratching.

7. Figure 4b is not clearly explained. What treatment was done to these mice to elicit spontaneous scratching?

8. Authors should make Figure 6 to figure 2. Current location of this figure is not appropriate.

9. In opto-tagging experiments, authors have found that all the recorded neutral cells are excited by the optical stimulation. This indicates both RVM ON and neutral cells express Tacr1. But authors do not acknowledge this in most of the study.

10. Current study shows role for RVM Tacr1 neurons in itch modulation, there is no direct evidence to show that its only via ON cells not via neutral cells or both.

[Editors' note: further revisions were suggested prior to acceptance, as described below.]

Thank you for resubmitting your work entitled "Inhibition of itch by neurokinin 1 receptor (Tacr1) -expressing ON cells in the rostral ventromedial medulla in mice" for further consideration by *eLife*. Your revised article has been evaluated by Kate Wassum (Senior Editor) and a Reviewing Editor.

The manuscript has been improved but there are some remaining issues that need to be addressed, as outlined below:

1. Owing to concerns about the quality of cFos staining, we recommend that these are deleted.

2. Revise the legend to Figure 2 and ensure that there are no inconsistencies with the text.

3. Correct textual errors as outlined by Review#1.

4. Specify the statistical analysis performed on results shown in Figure 4D.

5. Provide a correct reference for the text on line 59.

6. Please ensure your manuscript complies with the *eLife* policies for statistical reporting: https://reviewer.elifesciences.org/author-guide/full Report summary statistics (e.g., t, F values) and degrees of freedom, exact p-values, and 95% confidence intervals wherever possible. These should be reported for all key questions and not only when the p-value is less than 0.05.

*Reviewer #1 (Recommendations for the authors):*

The authors made a good faith attempt to validate the more ambitious claims from the original submission for which additional evidence was requested and was mostly unsuccessful. The manuscript is now more modest, yet still contributes significantly to our understanding of brainstem processing of scratching behavior in rodents. The main conclusion, that activation of Tacr1 expressing ON cells in the RVM reduces itch-evoked scratching, is supported by the data. The authors have also corrected some errors and improved some figures and clarified text. A number of careless errors still remain, however (listed below), and although the original hypothesis for a descending mechanism was unable to be supported by experimental evidence, the discussion still reaches this original conclusion with both hands.

Line 247 – "Importantly, our results support a role for these neurons in descending inhibition of acute and chronic itch." The results do not address descending inhibition.

It's a shame that a few more cells like in Figure 1 from the response to reviewers could not be found.

Here is a list of items that could be improved:

Figure 2: Legend and Results text: There are ambiguities regarding the percentages reported that should be written more clearly. For example: "(D) 84% of cells exhibited co-localization of tacr1 and mcherry" p32. Does this mean 84% of all cells in RVM? // "…there was significant increase in the number of c-fos positive cells following ip administration of clozapine…" p.6 – is this a percent of the total cells in RVM or a percent of mcherry-labelled cells? It is reported that the number of such cells is increased, but then percentages are reported and the ratios are not clearly defined.

Figure 1: Please include blue and red indicators for SP and vehicle in the actual figure.

3A and B are not mentioned in the results text.

Line 136 – says figure 4D, maybe is referring to 3D?

Line 137 – Figure 4E, maybe is referring to 3E?

Figure 4 (D) "Imiquimod induced significant increases in alloknesis scores on day 1, 3 and 5 of treatment in DREADDs (dark blue) and vector control (dark green) mice." Presumably, this was compared to day 0, I could not find statistics for this.

5D – results text states 10 Hz, but figure and figure legend say 5 Hz.

5E – the description of this in the results does not match the image in the panel. The panel appears to show a sustained light for 10ms and a burst of APs toward the end. Perhaps there is a better way to illustrate this experiment.

Maybe 6F refers to 5F?

5j is not mentioned in the results text.

Line 59 – I do not think Carstens 2020 is the ideal citation for "[GRPR cells] are considered essential for spinal itch transmission" – that claim and finding belong to the Chen lab and it would be most appropriate to cite their work here.

*Reviewer #3 (Recommendations for the authors):*

The revised manuscript by Follansbee et al. has addressed many of our previous concerns. The authors also revised the manuscript to reflect if the Tacr1 cells in the RVM are ON or neutral cells. The only concern I have is regarding the quality of the cFos staining that is demonstrated in the manuscript. cFos is a nuclear protein but the images the authors have provided do not reflect this. As cFos data does not change the outcome of the study, I suggest authors remove this data from the final manuscript for publication.

---

## [Author Response]

Essential revisions:1. Perform further experiments to show more convincingly that descending inhibition of itch signal in the spinal cord is the mechanism the tacr1 rvm 'on' neurons are using to reduce scratching. Consider direct functional evidence for descending modulation, as opposed to indirect spinal consequences of local modulation in the RVM, and report the proportion of the tacr1 rvm neurons that send descending projections to the spinal itch circuitry.

We thank the reviewers for this excellent comment. We attempted to perform several functional experiments to demonstrate that the descending component of RVM activation was necessary for inhibition of spinal itch transmission. We first attempted to record from lumbar itch sensitive neurons which received an AAV-DIO-ChR2-eYFP injection in the RVM of Tacr1-cre mice. Once an itch sensitive neuron was identified we applied 10 hz blue light stimulation to the spinal cord to selectively activate the descending fibers of RVM Tacr1 +/ChR2 + neurons (Author response imge 1). Unfortunately, technical difficulties limited the number of pruritogen-sensitive spinal neurons we were able to record. In the provided example we did observe a clear inhibition of chloroquine evoked activity during blue light stimulation which returned once the stimulation ended. However, given the low sample size this is only anecdotal which we will not include in the revision.

**Author response image 1. sa2fig1:** Response of spinal neuron to intradermal injection of chloroquine (CLQ) is suppressed by optogenetic stimulation in RVM. (a) diagram of experimental design. AAV5-DIO-ChR2-eYFP was injected into the RVM of Tacr1 cre mice. (B) A CLQ responsive neuron was identified and inhibition observed following blue light stimulation to the spinal cord to activate ChR2-expressing descending fibers.

We also performed behavioral experiments in which we injected a retrograde AAV (AAV-RetroDIO-ChR2-mCherry) into the spinal cord of Tacr1 cre mice and subsequently implanted an optic fiber which terminated into the RVM. Using this paradigm, we hoped to transfect TacR1 + neurons which project to the lumbar spinal cord including the RVM Tacr1 neurons, which would be selectively activated with optogenetic stimulation. Unfortunately, these mice did not show a consistent decrease in pruritogen evoked scratching during optogenetic stimulation (Autor response image 2). This is likely the result of poor efficaciousness in the retrograde labeling of RVM projecting neurons, which were sparse when verified histologically (Author response image 2,c).

Previous studies have reported that 31% of RVM ON-cells project to spinal cord (Venegas et al., 1984) and that 42.5% of NK-1R-expressing RVM cells project to spinal cord (Pinto et al., 2008). While we believe that descending modulation of itch-signaling spinal neurons is the most parsimonious explanation for our results, our additional experiments do not provide unequivocal support for this. Rather than performing even more experiments, we opt instead to change the language of the manuscript. We maintain that descending modulation is still a possibility but cannot exclude the possibility of a supraspinal mechanism being responsible for our observed effect.

**Author response image 2. sa2fig2:** Inability of optogenetic stimulation of spinally projecting RVM Tacr1 expressing neurons to suppress scratching behavior. (a) retrograde AAV was injected into the dorsal spinal cord of Tacr1 cre mice. An optic fiber was implanted into the RVM following viral injection. (b-c) Expression of mCherry was present, but weak, in the RVM. (d,e) Blue light stimulation did not significantly reduce histamine (d) or chloroquine (e) evoked scratching behavior. n = 6 group.

2. Include essential controls in experiments involving DREADDs

We have included the essential control experiments as it pertains to the DREADDs experiment. We compared ip saline versus clozapine in DREADDs expressing mice and found a significant increase in cfos positive cells following ip clozapine compared with ip saline. The results are provided in Author response image 3.

**Author response image 3. sa2fig3:** Figure 3: cfos expression in Tacr1 DREADDs expressing mice following i. p. Saline and i.p. Clozapine.

3. Address the caveat of heterogeneity of tacr1 cells in association with the claim of their identity as ON cells; accordingly either provide additional evidence or temper down claims.

We thank the reviewers for this comment. Most of the confusion relates to the 3 identified Neutral cells which were light sensitive.

However, given the absence of evidence that Neutral cells are involved in descending spinal modulation, we did not study this class further

and thus did not document what percentage of Neutral cells were light sensitive. We included these neutral cells to make the point that there are some cells in the RVM which express ChR2 but are not classified as ON cells. This could be the result of a small population of neurons which are not RVM ON cells but express Tacr1. We have included a description of this in the Discussion section.

4. Undertake changes in text and figures recommended by reviewers under the'recommendations' section of individual reviews, as appropriate.

Addressed individually below.

5. If you have not already done so, please ensure your manuscript complies with the eLife policies for statistical reporting: https://reviewer.elifesciences.org/author-guide/full Report exact p-values wherever possible alongside the summary statistics, degrees of freedom, and 95% confidence intervals. These should be reported for all key questions and not only when the p-value is less than 0.05.

We have provided a statistics page which includes the summary statistics, degrees of freedom, 95% confidence intervals, and p-values for each experiment.

6. Please also include a key resource table.

Included.

7. Please include housing details, including number of animals/cage (e.g., single v. pair v. group housed), light-dark cycle, cycle at test etc.

Included.

Reviewer #1 (Recommendations for the authors):Decades of penetrating investigation of nociception at the level of the brainstem has revealed a complex circuitry including ON and OFF cells that activate or deactivate, respectively, in response to noxious input. This manuscript begins to unwind the relationship of these cells and circuits to the signaling of itch. The main conclusion, that a population of Tacr1 expressing neurons in the rostroventral medulla are "ON" cells and that activation of these cells suppresses scratching is supported by anatomical, behavioral, and functional data. The strengths of these conclusions rest on the use of several approaches to manipulate activity (optogenetic, chemogenetic, and pharmacological) and from single unit recordings in anesthetized animals, which provide a high level of internal consistency. The use of appropriate blinding, detailed analysis of sex differences, and several experiments offering detail on the precision of optogenetic activation are other methodological strengths. While the data are strong, the interpretations at times stray into areas of less certainty, and a singular vision is presented where alternate possibilities still exist.One alternate possibility is that the activation of Tacr1 RVM neurons with substance P enhances nociception, and it is this enhanced pain that limits the scratching behavior, rather than a decrease in the itch sensation per se. This is supported by the lowered mechanical withdrawal threshold after DREADD activation of TacR1 neurons. Perhaps scratching becomes too painful, or perhaps the effectiveness of scratching is increased, and therefore fewer scratches are required for the same itch suppressing effect. Although Liu 2019 is cited in support (p. 20) of the authors' conclusions, there is not data in the present manuscript showing that the Tacr1 RVM ON cells under study directly modulate the spinal itch circuitry, or the GRPR cells.

We agree with the Reviewer’s last sentence and this has been addressed above. We agree that a pronociceptive effect of ON cells might reduce itch and mention this possibility; however there is currently limited evidence to support this idea.

The experiments in figure 2 are missing a control – DREADDs may have some constitutive activity, and therefore it is important to show with and without the DREADD activator CNO or clozapine (the authors use both interchangeably for some reason).

We thank the reviewer for their comments. Our c-fos data (original manuscript Figure 2e-I; revised manuscript Figure 2 h), does show that there is some baseline activity in the DREADDs expressing neurons without clozapine. This is not surprising since most RVM ON cells we recorded from (Figure 6 and Supp Figure 3) exhibited baseline firing and were transiently excited by the ip injection. That is why we chose to directly compare the DREADDs activation with a control vector group, to demonstrate an increase in activity in the DREADDs positive cells following injection of clozapine. Our behavioral experiments are intrinsically controlled through ip injection of clozapine versus saline in the same animal. We performed the suggested experiment and compared i.p. saline versus i.p. clozapine in our DREADDs mice to confirm (Figure 2 i). We observed a similar result and have updated figure 2 to reflect this change.

There is also the nagging problem that somehow motor systems are compromised by the activation of RVM neurons (the spinal cord appears to be fully labeled with mcherry Figure 4f, including the ventral horn). Although the contrasting findings between less scratching and more von Frey withdrawal suggest an intact system at the most basic level, more convincing would be showing enhanced scratching activity via inhibition of the Tacr1 RVM ON cells (a positive behavioral output). This would both put aside motor concerns and strengthen the functional role of these cells. Since according to the recordings in Figure 6c, ON these on cells have ongoing activity prior to the pinch stimulus, it should be possible to inhibit this tonic activity.

We thank the reviewer for this excellent suggestion. However, we did not perform the experiment with inhibitory DREADDs and rely on the observation of enhanced mechanical withdrawal as evidence against generalized motor suppression.

Figure 7 should include more of the findings from the results. No OFF cells are represented, lack of involvement of 5HT not shown.

We thank the reviewer for this suggestion. Given some of the comments from other reviewers we have decided to remove this summary figure due to limited direct evidence that RVM Tacr1 expressing neurons exert an effect on spinal pruriceptive transmission.

Line 78+ state briefly what prep is being used (anesthetized, in vivo)Indicate volume and concentration of drugs (ie substance P, hist, cq). This could even be in figure legend.

Added.

Figure 1 (also Figure 6) Please confirm accuracy of the Bregma coordinates.

Bregma coordinates shown in Figure 1d and 5j were confirmed based on anatomical landmarks and intramedullary injection of blue dye at the injection site (for Figure 1d) or microscopic identification of lesions made at the intramedullary recording site (Figure 5j).

Line 86: "enhancement of responses lasted 1 hr" please show.

This is shown in Figure 1C. The potentiated ON cell response persists for the 60 minute window following SP injection.

Figure 1 e: it looks like blue light is shining in this figure. Please use a different style for clarity.

We agree and have updated the figure with a light gray color.

Figure 2 – Are the figures a and e mixed up?

We thank the author for this observation. The panels were mixed up and have been corrected in this revised version.

Figure 2 – I couldn't understand the value of including i-k, seems duplicative.

In these experiments we compared mice which received a viral injection containing the excitatory DREADD (e-g) or a control vector containing only mCherry (i-k). This way we can compare the increased c-fos expression in DREADD (+) cells with DREADD (-) cells. To improve clarity and reduce duplication we have removed the analysis for measurement of mean intensity and present only the percentage of positive cells.

Figure 2 – legend "mean number of c-fos", but figure shows percentage, correct; also see line 119-121; again, this needs a no CNO control.

We have performed a saline (i.p.) control. We observe an increase in the number of cfos positive cells following clozapine (ip) when compared with saline in DREADDs mice (updated Figure 2 h).

Figure 3B would be more convincing for the authors' statement that "expression of hM3Dq-mcherry was limited to the RVM" if the area outside the RVM was included in the figure.

We apologize for the narrowness of the imaging window. We have updated Figure 3 b with a highquality coronal image showing the distribution of mCherry following DREADD injection. This shows that mCherry extends beyond the raphe nuclei into the adjacent areas that include nucleus gigantocellularis, which is included in Fields' definition of RVM.

Data between Figure 4 and Figure 3 are a bit different re: von Frey. Please explain the added value of the tac1rcre-er mice versus the previous mouse line and offer some insight into this discrepancy.

We thank the reviewer for this excellent observation. Figure 3 was the result established in our lab. Serendipitously, we found that our collaborating lab had performed a similar set of experiments which was consistent regarding the effect on itch related behavior (Figure 4). Since these two sets of experiments were conducted entirely independently, there were several differences in the experimental approach (Different Tacr1 cre lines, use of clozapine vs CNO, different labs, and different experimenters). We are not sure why the von Frey results differ between these two experiments, but speculate that it is related to the numerous differences in the experimental design. We have decided to move the results of our collaborators to the supplemental section to reduce confusion, since the results are largely duplicative. We wanted to include these results to highlight that this result was reproducible.

Figure 5 – this figure was awkward (particularly a and b). What is meant by the bars and stars on the side?

The bars and asterisks indicate the groups which were significantly different. We have included a modified version of this figure in the revised manuscript.

The shades of green are hard to distinguish.

We appreciate this comment. To better distinguish among the lines, we have increased the size of the symbols and have made the light green and light blue lines dashed. This helps the reader to differentiate the various lines and symbols.

The legend states "imiquimod…produced a significant increase in spontaneous scratching" – but I could not identify the statistical test used.

We have now included the statistical test used (paired t-tests) and correct the language used.

There is an indication that something is "red", but I see no red in the figure.

We apologize for this error. There are no red bars in the revised figure 4.

Why is clozapine used instead of CNO?

We used clozapine for all behavioral experiments except that shown in what is now Supplementary Figure 1 (conducted by the Pittsburgh laboratory, which used CNO). This is due to concerns mentioned in Gomez et al., 2017, which showed that CNO cannot cross the bloodbrain barrier, but instead breaks down into clozapine which can cross the blood-brain barrier and acts as a potent DREADDs agonist.

Why are imiquimod-evoked scratches "spontaneous"?

Imiquimod is applied ~23 hours prior to measurement of scratching behaviors, thus we call these scratches spontaneous since this is ongoing behavior.

Line 187 – "tip extended a few millimeters" This seems too far. I'm not sure it would work properly. The methods says micrometers.

We thank the reviewer for this observation. The electrode tip extended a few hundred microns past the end of the optic fiber. We have revised the manuscript.

Line 189 (and elsewhere), light used is given as less than 5mW. This is vague, and could be 0mW.

Agreed and we have revised.

Line 191 – "faithfully entrained 10Hz (6d)" but only 5 hz is shown.

This is a mistake and we have revised.

Line 219 – "doublet" action potentials. I was expecting a picture.

We have included an example in the supplemental figure 3d.

Line 227 – Figure 4 (should be supplemental).

Agreed.

Line 234 – reword; it sounds like the actual force of the limb withdrawal was greater.

Agreed, this has been corrected.

Line 246 – Please define what is the fev-cre line upon first introduction.

At the suggestion of the other reviewers we have removed our results pertaining to serotonergic neurons from this paper.

Line 305 – reference to Figure 4, but I think maybe Figure 2 is meant?

The reviewer is correct and this has been revised.

Reviewer #2 (Recommendations for the authors):Strengths: The results from this study broadly indicate that RVM TacR1-neurons can alter itch responses and are mainly ON-cells which are new findings that define important elements in the itch sensory system.Weaknesses: neurochemical experiments suggest that TacR1 is expressed in a heterogeneous population of RVM neurons. Further, the results from the use of two different mouse lines where TacR1-neurons are chemogenetically manipulated, differ from each other. These two issues make it difficult to completely unambiguously interpret results from chemogenetic experiments.Overall, the data in this manuscript is interesting and show important new findings. However, several of the interpretations made in the manuscript go beyond the results described and a revision of the manuscript is needed to reasonably reflect the presented results. There are also other questions about the Results and errors in the way that data is presented. First, the TacR1-neurons recorded from and manipulated are not directly shown to be descending. This is inferred from other studies and spinal cord staining but, what proportion of them are descending and could some of the effects reported be through indirect pathways? Further, the study does not directly establish that the effects seen occur in the spinal cord. For instance, some of the TacR1 neurons may be interneurons and some of the effects of activating TacR1 neurons may occur in other brain regions.

The Reviewer is correct and this comment is consistent with that of other Reviewers. For this reason, we have modified the text as described above.

Second, another concern is the incomplete characterization of the TacR1-cre mouse lines; the second TacR1-creERT2 mouse line does not seem to have been neurochemically characterized and the characterization of the one TacR1-cre line is not thorough (see comments below). Third, the parts of the manuscripts on 5HT signaling are not well connected with the other parts of the paper and need to be integrated or removed.

The section on serotonergic neurons has been removed as recommended.

Fourth, although it is not unreasonable to suggest that the OFF-cells are downstream of ON-cells, the manuscript does not provide direct proof of this; TTX, GABA inhibition experiments and measurements of latency responses relative to optogenetics are probably required to demonstrate this.

We agree.

I realize that all these requests would likely require considerable additional experimentation, and this may go beyond eLife guidelines. In the absence of performing new extra studies, the title, abstract and discussion will all need revisions to tone down conclusions to better reflect the data presented.

The revised text has toned down these conclusions.

Below is an itemized list of concerns:a) The data in figure1c is a bar graph, scatter plots should be used to allow a better assessment of the data.

We thank the reviewer for this suggestion and have revised our graphs accordingly.

b) Figure 1h, are the values on the y-axis correct? (8-2g, presumably 0.8g to 0.2g). This increase in sensitivity would be better described as a mild mechanical sensitization, similarly data in figure 3 should be described as mild-mechanical sensitization.

The force in grams is labeled correctly. This represents the instantaneous force measurements using the electronic con Frey apparatus. We agree with describing as mild mechanical sensitization and we have revised accordingly.

c) Figure 2, panels a and e have been switched. The staining for mCherry is different for each panel, what is the reason for this? – are they the same region or is A in the reticular region of the RVM and other images taken in the midline? I.e. were images taken from the same area of the RVM and if not, how was data normalized?

The reviewer is correct, the panels were inadvertently switched and we apologize for this error. We have corrected in this current version. Each panel (a, e, and i) represents a different experiment. Panel a represents mCherry expression in the NK-1R counter-staining experiments, panel e from our DREADDs injections and panel i from our control vector injections. Images were taken from approximately the same region of the RVM.

Since the RVM encompasses several separate brain nuclei, the authors should detail these, for instance; raphe magnus, raphe pallidus, PGRNL, and MARN.

We thank the reviewer for this suggestion. We have provided an updated coronal image with borders for the Gigantocellular, raphe magnus, and raphe pallidus nuclei. The RVM, as described by Howard Fields, includes the midline raphe and adjacent reticular formation extending into the gigantocellular nucleus. We see most of the expression within the raphe magnus/pallidus nuclei, and extending dorsally and laterally into the gigantocellular nucleus, consistent with this definition of RVM.

In addition, in the electrophysiology experiments described in figure 6, ON-cells and OFF-cells were probed in many areas of the general medulla – was the characterization of the lines used also similarly widely distributed?

Similar to our previous recordings on RVM ON and OFF cells (Follansbee et al., 2018), the recording sites were broadly distributed, often into the gigantoceullar nucleus. There was no apparent difference in ON/OFF cell response based on anatomical location within this area.

Images in figure 2 (or a new supplement) should be of whole coronal sections of the medulla in order to see the locations of all the TacR1-neurons in the medulla.

We generated a coronal section image which has been used in the revised version of the manuscript (Figure 3b).

d) Images in figure 2ab and c indicate 4 double positive and 4 TacR1 alone neurons, but the graph reports there is an 80% overlap. A move representative image should be presented.

The graph panels were out of order. This has been addressed in the revised version and is more representative.

e) Figure 2d, h, and l should be presented as scatter plots.

Changed.

f) Figure m, n, o same comment as d), an image showing more neurons would better illustrate the data presented in panel p.

We appreciate the reviewer’s suggestion, however we wanted to use representative images as taken from our analysis, which best correlated with the mean reported. These images are the full size. I took several images from each brain and averaged for each mouse.

g) Figure 2m shows endogenous TacR1 expression and does not report cells where cre mediated excision occurs. Therefore, this does not prove that cells chemogenetically manipulation or recorded from are GABAergic. The ISH should be on tissues from TacR1-cre mice infected with AAV-cherry and if different serotypes of AAV were used in different experiments this should be noted as a limitation of the study.

We have removed the section pertaining to GABA colocalization.

h) All numbers of counted neurons for figure 2 need to be reported.

Agreed and included.

i) Were similar neurochemical characterization studies performed on the second TacR1-creERT2 line? If they were not, then the authors should acknowledge that the differences in results between these lines may be because different populations of neurons might be activated.

Previous Figure 4 (now supplemental Figure 1) shows the only data obtained from the second Tacr1 line. However, this second line was not as completely evaluated. We wanted to include these data because the experiments were conducted entirely independently (in the Pittsburgh lab) and highlight the reproducibility of our behavioral scratching results.

j) Line 115, sentence needs rewording.

Agreed and fixed.

k) Line 129, this sentence repeats the previous one and is less accurate giving a false impression the presented data shows that all Tacr1-neurons are inhibitory. Similar inaccuracies are repeated throughout the manuscript.

We agree with the reviewer and have changed our language throughout the manuscript to be more explicit and not misleading.

l) Are the TacR1-neurons the same as those described in Francois et al. 2017?

Francois et al. drove expression in GABAergic cells, so any RVM cells which project to the spinal cord and express GABA would have been manipulated. Sarah Ross has evidence that OFF cells are also GABAergic, so some of the neurons described by Francois et al. may have been OFF cells. Our optotagging experiments suggest that OFF cells do not express Tacr1, while many ON cells do. Francois suggested they were activating RVM ON cells due to the predominant behavioral effect (mechanical sensitization) following activation of RVM spinally projecting GABAergic neurons. We have some limited evidence that RVM ON cells express GABA, thus it is possible that Francois activated Tacr1-expressing RVM ON cells, but the population of neurons they activated was likely more extensive and not limited to RVM ON cells.

m) Different serotypes of AAV are used in Figure 3 and 4, a comment on this should be added as this may additionally contribute to the differences in effects of chemogenetic manipulation between these experiments.

We completely agree. These experiments were conducted independently, without either investigator being aware the other was also conducting a similar experiment, so we weren’t able to coordinate these details.

n) 50ug of chloroquine produces different numbers of itch bouts in controls in figure 3 and 4, what's the reason?

These experiments were performed by completely different investigators, independently, and in a different line of mice.

o) Given that these experiments are suggested to confirm each other, is there a reason why in figure 3 von Frey thresholds are not affected but in figure 4 they are? This difference needs to be acknowledged in interpreting these results.

The reviewer brings up a great point. These two experiments were conducted entirely independently, without knowledge of each other. Figure 3, was conducted by TF in the Carstens Lab while (Previous version Figure 4) was conducted by EN in the Ross lab. Thus, there were numerous differences including: Lab location, mouse line, virus serotype, behaviorist, behavior apparatus, DREADDs actuator (clozapine vs CNO), etc. We found the similarity of our results astounding and included both sets of data as validation of the reproducibility of our results. However, this led to confusion and we decided to put the secondary results from the Ross lab (Previous Figure 4) in the supplemental section and focus primarily on the results generated from the Carstens lab.

p) Figure 4e, the staining for mCherry seems to extend beyond the cell bodies of neurons in the raphe magnus (dorsal and lateral), are these dendrites and could the AAV retrogradely infect, in the RVM, terminals of neurons in other brain regions? A larger image would help resolve whether the staining is from dendrites.

The expression of mCherry is not limited to the cell bodies and we frequently see strong expression of mCherry throughout the entire neuron (mainly soma and dendritic field), but additionally in the axons. In our injections, we see strong expression of mCherry at the injection site in the RVM, but not a lot of expression elsewhere. A lower-power image of the brainstem has been included in revised figure 3b.

q) Figure 4f, the contrast used for this image makes it hard to see any staining. Is there bilateral staining and which lamina are these terminals projecting to and are projections the same in all spinal cord segments?

We agree and this image has been removed from the manuscript.

r) The electrophysiology of optogenetic responses of ON and OFF-cells- optotagging. What were the waveforms and latency of the proposed directly activated versus indirectly activated neurons and were there any indications there is more than one class of ON-cells?

We did not observe any striking difference in spike waveforms between different cell types. There was no indication of more than one class of ON cell.

Was supervised/unsupervised clustering used to distinguish cell-types?

No.

Even though the evoked response reliability of ON-cells is high, the latency of these neurons in slow compared to other optotagging studies reported.

The latency from optotagged neurons varies drastically across cell type and light intensity. We found it necessary to use latency, efficiency index and firing rate to determine whether the cell was responding directly. Another study of hippocampal cells (Zhang et al., 2013) reported latencies of similar duration (~9 msec) compared to our present data.

What is the response reliability of OFF-cells is the distribution of latency for ON-cells suggest more than one class?

Sorry, we don’t understand the question.

And what's the average latency of response of OFF-cells?

No OFF cells we recorded from responded with excitation to the light stimulus, but 6/12 responded with inhibition to the light stimulus.

Were experiments with NBQX and bicuculine performed to show direct activation?

No, we did not try to block synaptic transmission in these experiments because this paper was largely focused on Tacr1 expressing ON cells. But we thought the OFF cell response was of interest and wanted to include the observation in this manuscript. We did not try applying NBQX or bicuculine because it was too difficult to construct a microinjection/optic fiber/microelectrode device which would be required for these experiments and thus was beyond the scope of the present study.

s) Figure 6b shows a small area of where recordings were made (versus panel j). A larger view would be good.

We thank the reviewer for this comment. Unfortunately, at the time of these recordings, we were limited in our microscopy experiments and could only image at 20-40x. The injections for these experiments are quite similar to the injections made for our chemogenetic experiments (same injection coordinates and virus serotype). Figure 5b is not a lesion site but demonstration of expression of YFP, and thus ChR2.

t) Were studies performed to show ON-cells are synaptically connected to OFF-cells? I.e. for OFF-cells could responses be blocked with TTX, or since the TacR1-cells are suggested to be mainly GABAergic are these responses blocked with bicuculline? Without these results, it is possible that TacR1 is expressed in both ON and OFF-cells.

As mentioned above, RVM OFF cells were never excited by light, but were instead inhibited by optic stimulation. While Tacr1 expressing cells might be heterogenous, we have no evidence to suggest RVM OFF cells express Tacr1.

u) The section on sex differences is odd. It needs to be integrated with the other parts of the paper.

We agree with the reviewer and have integrated the sex differences in the revised manuscript.

v) Section of 5HT RVM cells does not connect with other parts of the paper and basic control data are not presented for these studies. I would recommend this section be removed as it does not add to the main theme of the paper and the data is too preliminary.

We agree and have removed this from the manuscript.

Reviewer #3 (Recommendations for the authors):The manuscript looks at contributory role of a RVM Tacr1-expressing neurons in the acute and chronic pruritis. Authors demonstrate that RVM ON and neutral cells express Tacr1 and send projections to the spinal cord, implicating possible role in descending inhibition of spinal pruriceptive transmission. Overall, the study used well established and complementary methods to support their findings. This work fills significant gaps in the descending itch literature. Largely I think the paper is well presented, data supports the conclusions of the study. There are some concerns regarding the figures. The manuscript could improve if authors address these concerns.1. Authors show that in Figure 1, SP injections into the RVM significantly reduced evoked itch and von Frey behaviors. It is confusing as the data is presented, authors say they are using Paired student test. But in the methods, it is not clear if they use same mice for saline and SP injections. Paired T test would be appropriate if pre and post were conducted on same mice.

We apologize for the confusion. In these experiments the same mice received both saline and SP and were analyzed using a paired t test.

2. SP microinjections did not always decrease evoked itch behaviors. Any differences observed in the location of injections that explains this variance in the data?

This is a distinct possibility, although we did not observe a correlation between injection location and the degree of suppression of scratching. Itch behavior is notoriously variable. Especially for saline injections that produce low initial scratch bouts, it could be a floor effect. Nonetheless, the overall decrease is significant and consistent across several lines of evidence.

3. Figure 2a and 2e seemed to be switched. Authors should address this.

Thank you for your observation. We have fixed this issue.

4. cFos is a nuclear protein. In the images authors show, it seems like they are just capturing background autofluorescence instead of cFos signal.

We thank the reviewer for this astute comment. We only measured cfos expression in mCherry positive cells. The acquisition settings were maintained constant for DREADDs and Control vector imaging and apologize for the bright image provided. In addition to the measurement of mean intensity, we also used the traditional method of visually counting cfos positive cells and achieved similar numbers.

5. Authors claim Tacr1 neurons are Vgat+ve. Could authors comment on if they have looked to see any Tacr1 neurons are Vglut positive. It is possible because it has not been tested there could be neurons Tacr1+ve neurons that could be Vglut+ve.

We have decided to remove the section colocalizing Tacr1 and GABA.

6. In Figure 3, authors seem to use same panels for histamine and chloroquine evoked scratching.

We thank the reviewer for this astute observation and apologize for this inadvertent error. We have replaced the panel showing chloroquine-evoked scratching behavior. The conclusions remain the same.

7. Figure 4b is not clearly explained. What treatment was done to these mice to elicit spontaneous scratching?

The spontaneous scratching observed was with no treatment. The mice used for this behavior demonstrated some amount of “baseline” scratching bouts, which were reduced during administration of CNO.

8. Authors should make Figure 6 to figure 2. Current location of this figure is not appropriate.

We thank the reviewer for this suggestion. We ultimately decided to remove this figure at the suggestion of other reviewers.

9. In opto-tagging experiments, authors have found that all the recorded neutral cells are excited by the optical stimulation. This indicates both RVM ON and neutral cells express Tacr1. But authors do not acknowledge this in most of the study.

We apologize for the confusion. This study did not include a comprehensive investigation of Neutral cells. Qualitatively speaking, the large majority of Neutral cells were not light sensitive, but I felt it important to acknowledge that some (n=3) were light sensitive. For example, in the neutral cells we recorded from, there was no baseline activity and they do not respond to nociceptive stimuli, so it would be impossible to record from similar cells if it weren’t for the occasional light sensitive neurons. We agree with the reviewer that we should explicitly mention the heterogeneity of Tacr1 expressing cells and have revised the manuscript to reflect this.

10. Current study shows role for RVM Tacr1 neurons in itch modulation, there is no direct evidence to show that its only via ON cells not via neutral cells or both.

We agree that our conclusions are on RVM Tacr1 expressing cells. However, our evidence, suggests that this is likely overlapping with RVM ON cells. The reviewer makes a good point, but we are reluctant to suggest a role for Neutral cells in descending itch modulation since their firing does not change in response to noxious stimulation.

[Editors' note: further revisions were suggested prior to acceptance, as described below.]

The manuscript has been improved but there are some remaining issues that need to be addressed, as outlined below:1. Owing to concerns about the quality of cFos staining, we recommend that these are deleted.

We thank the editors and reviewers for this suggestion. We have removed the cfos experiments from the revised manuscript. Perhaps the cfos data could be included as a supplemental figure?

2. Revise the legend to Figure 2 and ensure that there are no inconsistencies with the text.

In addition to removing the cfos data from this figure, we have clarified the analysis used in the text and in the figure legend.

3. Correct textual errors as outlined by Review#1.

We have made the changes requested by Reviewer #1 (see individual points below).

4. Specify the statistical analysis performed on results shown in Figure 4D.

We have added this information to the figure legend as well as our summary statistics file.

5. Provide a correct reference for the text on line 59.6. Please ensure your manuscript complies with the eLife policies for statistical reporting: https://reviewer.elifesciences.org/author-guide/full Report summary statistics (e.g., t, F values) and degrees of freedom, exact p-values, and 95% confidence intervals wherever possible. These should be reported for all key questions and not only when the p-value is less than 0.05.

We thank the editor for this reminder. In the author guide it states: “In some cases, it might be unwieldy to have this information in the legend of a figure, in which case the information can be provided in a source data file”. We have opted to include our statistics as source data files associated with each figure, which include the summary statistics and the raw data. We hope that this is satisfactory. If not, please let me know and I will add the statistics to the figure legends.

Reviewer #1 (Recommendations for the authors):The authors made a good faith attempt to validate the more ambitious claims from the original submission for which additional evidence was requested and was mostly unsuccessful. The manuscript is now more modest, yet still contributes significantly to our understanding of brainstem processing of scratching behavior in rodents. The main conclusion, that activation of Tacr1 expressing ON cells in the RVM reduces itch-evoked scratching, is supported by the data. The authors have also corrected some errors and improved some figures and clarified text. A number of careless errors still remain, however (listed below), and although the original hypothesis for a descending mechanism was unable to be supported by experimental evidence, the discussion still reaches this original conclusion with both hands.

We thank the reviewer for their comments and careful review of our manuscript.

Line 247 – "Importantly, our results support a role for these neurons in descending inhibition of acute and chronic itch." The results do not address descending inhibition.It's a shame that a few more cells like in Figure 1 from the response to reviewers could not be found.

We agree with the reviewer and are likewise disappointed that we could not find more neurons in an appropriate amount of time.

Here is a list of items that could be improved:Figure 2: Legend and Results text: There are ambiguities regarding the percentages reported that should be written more clearly. For example: "(D) 84% of cells exhibited co-localization of tacr1 and mcherry" p32. Does this mean 84% of all cells in RVM?

We apologize for the ambiguity. For this analysis we evaluated Tacr1 expression specifically in the mCherry-expressing population. The result is the percentage of mCherry expressing neurons which also express Tacr1 and not all cells in the RVM. We have added clarifying statements in the figure legend as well as in the methods under Immunofluorescence.

// "…there was significant increase in the number of c-fos positive cells following ip administration of clozapine…" p.6 – is this a percent of the total cells in RVM or a percent of mcherry-labelled cells? It is reported that the number of such cells is increased, but then percentages are reported and the ratios are not clearly defined.

Again, we apologize for any ambiguity in our writing. At the behest of another reviewer, we have decided to remove the cfos labeling experiments.

Figure 1: Please include blue and red indicators for SP and vehicle in the actual figure.

Fixed.

3A and B are not mentioned in the results text.

Added.

Line 136 – says figure 4D, maybe is referring to 3D?

Corrected.

Line 137 – Figure 4E, maybe is referring to 3E?

Corrected.

Figure 4 (D) "Imiquimod induced significant increases in alloknesis scores on day 1, 3 and 5 of treatment in DREADDs (dark blue) and vector control (dark green) mice." Presumably, this was compared to day 0, I could not find statistics for this.

The reviewer is correct. For this statement, we conducted a 1 way ANOVA and compared days 1, 3, and 5 with Day 0. We will add these details into the figure legend and add the analysis to our statistics materials.

5D – results text states 10 Hz, but figure and figure legend say 5 Hz.

Fixed. The correct frequency is 5 Hz.

5E – the description of this in the results does not match the image in the panel. The panel appears to show a sustained light for 10ms and a burst of APs toward the end. Perhaps there is a better way to illustrate this experiment.

The results shown here are shown as a peristimulus time histogram (PSTH). For all the stimulation events (light pulses) applied at 2 Hz for this neuron, the action potentials within a 20 ms period are summated. This shows the average latency to respond as well as the jitter of response. It is our belief that this graph is informative and important to convey the consistency of this neuron to respond to the optic stimulation.

Maybe 6F refers to 5F?

Fixed.

5j is not mentioned in the results text.

Added.

Line 59 – I do not think Carstens 2020 is the ideal citation for "[GRPR cells] are considered essential for spinal itch transmission" – that claim and finding belong to the Chen lab and it would be most appropriate to cite their work here.

Agreed.

Reviewer #3 (Recommendations for the authors):The revised manuscript by Follansbee et al. has addressed many of our previous concerns. The authors also revised the manuscript to reflect if the Tacr1 cells in the RVM are ON or neutral cells. The only concern I have is regarding the quality of the cFos staining that is demonstrated in the manuscript. cFos is a nuclear protein but the images the authors have provided do not reflect this. As cFos data does not change the outcome of the study, I suggest authors remove this data from the final manuscript for publication.

We thank the reviewer for this observation. Given the concerns about the specificity of the cfos labeling we have removed this data from the revised manuscript.